# Emergence of the London Millennium Bridge instability without synchronisation

Igor Belykh [1,2 ✉], Mateusz Bocian [3,4], Alan R. Champneys[5], Kevin Daley [1], Russell Jeter[1], John H. G. Macdonald [6] & Allan McRobie[7]

The pedestrian-induced instability of the London Millennium Bridge is a widely used example of Kuramoto synchronisation. Yet, reviewing observational, experimental, and modelling evidence, we argue that increased coherence of pedestrians' foot placement is a consequence of, not a cause of the instability. Instead, uncorrelated pedestrians produce positive feedback, through negative damping on average, that can initiate significant lateral bridge vibration over a wide range of natural frequencies. We present a simple general formula that quantifies this effect, and illustrate it through simulation of three mathematical models, including one with strong propensity for synchronisation. Despite subtle effects of gait strategies in determining precise instability thresholds, our results show that average negative damping is always the trigger. More broadly, we describe an alternative to Kuramoto theory for emergence of coherent oscillations in nature; collective contributions from incoherent agents need not cancel, but can provide positive feedback on average, leading to global limit-cycle motion.

[1] Department of Mathematics and Statistics, Georgia State University, Atlanta, GA, USA. [2] Department of Control Theory, Lobachevsky University, Nizhny Novgorod, Russia. [3] Department of Roads, Bridges, Railways and Airports, Wrocław University of Science and Technology, Wrocław, Poland. [4] School of Engineering, University of Leicester, Leicester, UK. [5] Department of Engineering Mathematics, University of Bristol, Bristol, UK. [6] Department of Civil Engineering, University of Bristol, Bristol, UK. [7] Department of Engineering, University of Cambridge, Cambridge, UK. ✉email: ibelykh@gsu.edu

Synchronisation of coupled near-identical oscillators leads to emergent order in both natural and engineered complex systems[1–8]. The theory of weakly coupled near-identical oscillators, due to Kuramoto[9,10], has proved remarkably successful in explaining these phenomena. The pedestrian-induced instability on the opening day of the London Millennium Bridge[11] is often used as the canonical example; a threshold number of walkers enabled them to synchronise their footsteps with each other at a bridge natural vibration frequency[12]. However, since then, a number of publications have cast doubt on this explanation[13–18]. Yet, the explanation that Kuramoto-style synchronisation of the phase of walkers' foot placements remains part of the scientific zeitgeist[19].

In this work, we propose an alternative theory by arguing strongly for the more likely explanation that any synchronisation of pedestrians' foot placement is a consequence of, not a cause of the instability; a result that is consistent with observations on almost 30 bridges. Instead, we explain how uncorrelated pedestrians produce negative lateral damping on average to initiate significant bridge vibration, over a range of bridge natural frequencies. We present a simple formula that quantifies the effective total negative damping per pedestrian, and the contributions towards it from three distinct effects. We also show how this formula predicts the critical number of pedestrians in three distinct simulation models, including one that has a strong propensity for synchronisation[20]. The models also point to an almost universal frequency dependence of the instability criterion. More broadly than implications on design criteria for safe human-structure interaction, our work points to an alternative mechanism for emergence of collective behaviour in complex systems.

Kuramoto-like synchronisation analysis has so-far been unable to explain many of the instability features observed on the London Millennium Bridge and many other bridges (see Tables 1 and 2 below for a complete summary of known observations). The main features of this instability are: (1) bridges can exhibit large vibration amplitudes in more than one mode of vibration simultaneously, which need not be tuned to a particular walking frequency[13,21]; (2) a critical number of pedestrians is required in order to cause an instability[22,23]; (3) evidence of pedestrian footstep synchronisation[8,24] is scant, with the most definitive study estimating only 20% of the crowd walked in time with the bridge motion[25]; (4) engineering consultants Arup, who re-engineered the London Millennium Bridge, found that each pedestrian added, on average, effective negative damping[22]; retrofitting additional dampers successfully cures the problem[26].

One of the first to call into question the synchronisation explanation of the London Millennium Bridge instability was Nobel prize winner Brian Josephson, writing four days after the bridge's opening[27]:

> "The Millennium Bridge problem has little to do with crowds walking in step: it is connected with what people do as they try to maintain balance if the surface on which they are walking starts to move, and is similar to what can happen if a number of people stand up at the same time in a small boat. It is possible in both cases that the movements that people make as they try to maintain their balance lead to an increase in whatever swaying is already present, so that the swaying goes on getting worse."

Intuitive reasoning, underlying Josephson's argument and Arup's observations, suggests that to retain balance, each pedestrian should seek to lose angular momentum within their frontal plane. Further, Barker[28] identified a stepping mechanism whereby forces to the left and right do not necessarily average out. Therefore, on average, lateral vibration energy is transferred from the pedestrian to the bridge vibration mode. In effect, each pedestrian applies negative damping to the bridge.

In fact, the situation is more subtle. The interaction force at the bridge vibration frequency can be decomposed into components in phase with the bridge's acceleration and in phase with the bridge's velocity. The former changes the effective inertia of the bridge motion, whereas the latter changes the bridge's effective damping[29,30]. Paradoxically, for some specific combinations of the bridge vibration and pedestrian walking frequencies, a theoretical argument suggests[18,31] that the pedestrian can effectively extract energy from the bridge, which has been confirmed in laboratory treadmill tests[15,32,33].

Until now, it has been hard to quantify this negative damping effect in a model-independent way. A number of theories have been proposed for its physical origin[17,18,28,31]; however, it is not clear whether negative damping can be a consequence of synchronisation[34] or vice versa.

In this paper, we provide a compelling answer to this question in a multi-pronged approach: a comprehensive review of observational evidence; a new model-independent expression for the average negative damping effect; a detailed explanation of how negative damping is a natural consequence of pedestrian motion on average; simulation studies of several simple models for bridge-deck interaction; a careful explanation of the subtlety of the problem, for example, on the frequency dependence of the negative-damping effect and how synchronisation (or more precisely, coherence) of foot placements can have either an accentuating or moderating effect on the underlying instability. Further details are presented in "Methods" and in the Supplementary Information. We point to a broader scientific lesson of the London Millennium Bridge story: there is an emergent instability with an underlying frequency that can be excited by the uncorrelated behaviour of individual agents, who do not need to act in a coordinated manner. We suggest that such a paradigm may be helpful to explain other emergent oscillatory phenomena that have previously been ascribed to Kuramoto-style synchronisation; specifically the emergence of global economic cycles and the coordinated response of tiny hair-like structures within animal hearing organs.

## Results

**Review of observational and experimental evidence.** When crossing a bridge, most people take for granted that the bridge will remain steady and support them, but history shows that this is not always the case. The first documented pedestrian bridge incident dates back to April 12, 1831 when one of Europe's first suspension bridges, England's Broughton Suspension Bridge, collapsed due to dynamical instability induced by marching troops. The prevailing wisdom since is that soldiers should avoid marching in step, in case their stepping frequency might resonate with a natural (vertical) vibration frequency of the bridge. It is now established practice that soldiers are given the command to "break step" upon crossing a bridge to avoid just such a phenomenon. Vertical vibrations of bridges due to random excitation from pedestrians are still of concern, but prior to the year 2000 lateral vibrations were given little attention. This was because, for normal walking, the lateral component of the ground reaction force is an order of magnitude smaller than the vertical component and in the absence of coherence between pedestrians the resulting bridge responses were assumed to be negligible.

The London Millennium Bridge was designed as a collaboration between engineers, architects, and artists, as a very low profile suspension bridge. Without visually intrusive vertical cables, the intention was that the structure would appear from the side to be like a mysterious long blade, spanning the river with little visible support. The unusual geometry of the slender span contributed to the bridge having greater flexibility than most

**Table 1 Reported cases of lateral bridge instability due to the action of walking pedestrians. The final column documents any evidence presented for pedestrian synchronisation.**

| Bridge | Country | Year reported | Bridge type | Length (m) | Frequency (Hz) | Observation | Sync evidence |
|---|---|---|---|---|---|---|---|
| Erlach Footbridge[63] | Germany | 1972 | Several span continuous girder; main span supported by arch | 110 | 1.12 | Strong response with 300–400 crossing pedestrians | No evidence |
| Toda Park Bridge[40] | Japan | 1993 | Cable stayed; steel box-girder deck | 179 | 0.9 | ≤2000 pedestrians (2.1 ped/m²; amplitude in excess of 0.01 m; increase of vibration frequency during moderate occupancy | ≤20% synchronised pedestrians estimated from video analysis |
| Léopold-Sédar-Senghor Footbridge[64] | France | 1999 | Shallow steel arch | 140 | 0.81 | Exponential growth once amplitude reached 0.1–0.15 m/s² | No evidence |
| London Millennium Bridge[11,22] | UK | 2000 | Shallow suspension | 325 | 0.5, 0.8, 1.0 | 1.3–1.5 ped/m²; 1.86–2.45 m/s² max acceleration; pedestrians alternately tuned and detuned their pace with lateral bridge motion | No direct evidence; vertical pedestrian force random while lateral force correlated with bridge motion |
| Lardal Footbridge[65] | Norway | 2001 | Shallow glue-laminated timber arch | 91 and two approach spans of 13 | 0.83 | >1 m/s² for 40 pedestrians | No evidence; evidence of saturation (self-limiting) effect |
| Maple Valley Great Suspension Bridge[66] | Japan | 2002 | Suspension | 440 | 0.88, 1.02 | 0.045 m max displacement (1.35 m/s²); 0.7–1.3 ped/m² | Frequency synchronisation and "tuned and not tuned" effect from accelerometers on pedestrians' waists |
| Geneva Airport Footbridge[67] | Switzerland | 2002 | Reinforced concrete multi-span | 94.5 | 1.0 | One-directional traffic; "bordered on panic" while rapidly evacuating bridge | No evidence |
| Changi Mezzanine Bridge[21] | Singapore | 2002 | Shallow steel arch | 140 | 0.9 | 0.055 m (0.17 m/s²) | No evidence |
| Clifton Suspension Bridge[13] | UK | 2003 | Suspension | 214 | 0.53, 0.77 | 1.1 ped/m²; max 0.2 m/s² = 0.011 m | Evidence of a lack of synchronisation |
| Pedro and Ines Footbridge[23] | Portugal | 2006 | Multispan with shall steel main arch | 275 | 0.91 | Abrupt amplitude increase once critical number of pedestrian reached; max 0.2 m/s² for 73 ped and 1.2 m/s² = 0.04 m for 145 ped | No evidence |
| Simone de Beauvoir Footbridge[68] | France | 2006 | Shallow arch with tension links | 304 | 0.56, 1.12 | 0.03 m for 80–100 pedestrians with 20 synced; 0.06 m for 60 synced pedestrians | Tests with imposed synchrony showed saturation effect |
| Cragside Bridge[69] | UK | 2006 | Wrought iron arch | 69 | 2.8 | Increase of vibration frequency during pedestrian loading; max. acceleration amplitude 13.9 m/s² for 9 pedestrians walking at 110 steps/min | Tested under intentional synchronisation |
| Weil-am-Rhein Footbridge[70] | Switzerland | 2007 | Arch | 230 | 0.95 | 1.7 m/s² = approx. 0.08 m peak-to-peak with 800 people | Limited tuning effect during crowd load testing and argued to propagate in the crowd |
| Squibb Park Bridge[38,71] | USA | 2013 | Underslung suspension | 122 | 0.84 | N/A | N/A |
| Luiz I Bridge[72] | Portugal | 2020 | Double-deck metallic truss incorporating parabolic arch | 391.5, 172 | 0.73, 0.95 | Instability can be triggered independently at two vibration modes | No evidence |

bridges in the lateral direction, giving natural frequencies similar to typical pedestrian stride frequencies, while its relatively low mass also made it susceptible to significant vibrations. There is a widely available video that shows dozens of people rocking from side to side on the London Millennium Bridge's opening day, seemingly in time with the bridge, which is often used as compelling evidence for pedestrian synchronisation in popular media[19]. However, we encourage the reader to look again. A distinction needs to be made between synchronisation of head and upper body movements (readily seen in videos) and synchronisation of footfalls on the deck. We are not aware of any video footage that establishes that footfall synchrony occurred. Indeed, there is possible evidence in that video of lack of footfall synchrony, because pedestrian forward velocities vary widely. Moreover, a walker providing an effective negative damping force to the bridge, necessarily at the bridge frequency, will exhibit a component of upper body motion at that frequency.

In fact this same phenomenon of a lateral instability of pedestrian bridges had been seen before, and there is evidence going back to 1972. The complete list of pedestrian bridges that are known to have developed lateral oscillation due to pedestrian motion runs to at least 30 separate examples; see Table 1 for a list of those for which there are detailed scientific reports and Table 2 for others for which quantitative evidence is not available. Note in the final column of these tables the scant evidence for pedestrian synchronisation being observed.

The geography of such crowd-induced instability events is truly worldwide. It includes the massive Bosphorus Bridge linking Asia and Europe[35] and an icon of Lower Manhattan, the Brooklyn Bridge which started swaying as a crowd of pedestrians trudged across during the 2003 blackout. When packed shoulder to shoulder with pedestrians, the bridge started vibrating making pedestrians lose balance and feel seasick[36]. The Brooklyn Bridge repeatedly experienced crowd-induced instabilities during the 2011 protest and 2011 New Year's celebration[34] raising the concern that "Manhattans's emergency exit"—as the bridge is sometimes called—is not built for crowds.

Coincidentally, one of the more recent examples of lateral pedestrian instabilities is Squibb Park Bridge, also in Brooklyn (it is a city of bridges, after all)[37]. Opened in 2013, this $3.9-million wooden park bridge was purposefully designed to bounce lightly but over time the increased bouncing and lateral swaying became a safety concern for pedestrians[38]. Three years after it was initially closed for $2.5-million repairs, the Squibb Park Bridge reopened in April 2017[39] but was later demolished in 2019 amid concerns of its structural integrity.

While the evidence of bridge instabilities is often anecdotal, some direct measurements of bridge response characteristics are available for recent crowd-induced instability events involving the Toda Park Bridge in Japan[40], Solférino Bridge in Paris[41], the London Millennium Bridge[22], the Maple Valley Great Suspension Bridge in Japan[25], Singapore Airport's Changi Mezzanine Bridge[21], the Clifton Suspension Bridge in Bristol, UK[13], and the Pedro e Inês Footbridge in Portugal[23].

A particularly notable observation was the instability due to crowds returning from an annual hot-air balloon festival across Bristol's iconic Clifton Suspension Bridge[13]. Since vibrations of the bridge had been observed during previous crowd events, Macdonald was commissioned by the bridge's operating trust to fit accelerometers to record the vibrations as the instability occurred. Observations showed that two lateral modes of vibration were excited simultaneously by the large pedestrian crowd, neither of which was tuned to the average walking frequency. Since then, the trust has stipulated that the bridge must remain closed to all pedestrians and other traffic at peak times during the balloon festival.

**Analytical prediction**. We have established a general expression for the average contribution to the bridge damping of the interaction force of a single pedestrian over one gait cycle. We have found that this increment $\sigma$ can be written as the sum of three components (see Methods):

$\sigma_1$  coefficient of lateral bridge velocity-dependent component of pedestrian foot force on bridge, ignoring gait timing adjustment,

$\sigma_2$  coefficient of lateral bridge velocity-dependent component of force due to adjustment of pedestrian lateral gait timing, and

$\sigma_3$  coefficient of lateral bridge velocity-dependent component of force due to adjustment to forward gait .

The terms $\sigma_2$ and $\sigma_3$ depend on the timing of stepping behaviour of pedestrians in response to the bridge motion. However, in all our simulations, we have found $\sigma_1$ to be the most important effect in triggering large-amplitude vibrations (see the Supplementary Information). This effect is perhaps counter-intuitive, since it may be imagined that, in the absence of phase synchrony between the bridge and pedestrian, the lateral foot force on the bridge would average to zero. However, this is not the case; see Fig. 1 for a detailed explanation.

The expressions for $\sigma_1-\sigma_3$ should be evaluated individually for each pedestrian $i$ and will depend on that pedestrian's stride frequency $\omega_i$ as well as the vibration frequency $\Omega$ of the bridge in the mode in question. Thus, we can write the total effective damping coefficient $c_T$ of the bridge with $N$ pedestrians as

$$c_T = c_0 + N\overline{\sigma}(\overline{\omega},\Omega) := c_0 + \sum_{i=1}^{N}\left(\sigma_1^{(i)}(\omega_i,\Omega) + \sigma_2^{(i)}(\omega_i,\Omega) + \sigma_3^{(i)}(\omega_i,\Omega)\right),$$

(1)

where $c_0$ is the coefficient of natural (passive) damping of the bridge, $\overline{\sigma}(\overline{\omega},\Omega)$ is the average damping coefficient per pedestrian, and $\overline{\omega}$ represents the mean pedestrian stride frequency.

We have found, over large ranges of pedestrian and bridge frequencies, that $\overline{\sigma} < 0$ on average. Imagine a thought experiment in which pedestrians are added to a bridge deck one by one, then when we reach a critical number

$$N = N_{\text{crit}} = -c_0/\overline{\sigma}$$

(2)

of pedestrians, the overall modal damping $c_T$ of the bridge will become negative. Negative damping will cause the amplitude of the bridge vibration mode to grow exponentially.

**Simulation results**. To test this theory we have performed simulations on three different mathematical models describing a number of pedestrians coupled with a lateral bridge mode (see Methods for model descriptions). In each case we take a parsimonious assumption, justified in the relevant literature, that walking is fundamentally a process in which the stance leg acts as a rigid strut, causing the body centre of mass (CoM) to act like an inverted pendulum in the frontal plane[18,31,42,43] during each footstep. Rather than fall over, the step ends when the other leg strikes the ground and, ignoring the brief double-stance phase seen in realistic gaits, the pedestrian switches to an inverted pendulum on that leg. We consider a single lateral vibration mode of the bridge, forced by the motion of $N$ pedestrians walking in a direction perpendicular to this vibration. Any interaction between pedestrians other than indirectly through the bridge motion is ignored.

**Table 2 Other reported instances of lateral pedestrian-induced bridge vibrations.**

| Bridge | Country | Year | Observation |
|---|---|---|---|
| Angers Bridge[73] | France | 1850 | Collapsed while a battalion of soldiers was marching across the bridge, killing 226 of them; the bridge movement "involuntarily gave the soldier a certain cadence" |
| Brooklyn Bridge[74] | USA | 1880 | Swaying of catwalks during construction |
| Wuhan Yangtze Bridge[75] | China | 1957 | |
| Kiev suspension bridge[76] | Ukraine | 1958 | |
| Bosporus Bridge, Istanbul[35] | Turkey | 1973 | 100,000 pedestrians on opening day caused it to sway |
| Auckland Harbour Bridge[22] | New Zealand | 1975 | 0.67 Hz oscillation during public demonstration |
| Groves Bridge, Chester[22] | UK | 1977 | 100 m suspension bridge filled with rowing regatta spectators |
| Golden Gate Bridge[77] | USA | 1987 | Oscillations occurred due to a crowd of pedestrians crossing the bridge to mark the bridge's 50th opening anniversary |
| NEC, Birmingham[22] | UK | 1990 | 0.7 Hz oscillations of 45 m bridge linking exhibition centre to railway station after major events |
| Expo 1998 footbridges, Lisbon[78] | Portugal | 1998 | "Acceleration in horizontal vibrations can go over adequate limits with just a few pedestrians." |
| Alexandra Bridge, Ottawa[11] | Canada | 2000 | Crowd due to firework display |
| Brooklyn Bridge[36] | USA | 2003 | "Packed shoulder to shoulder with pedestrians" during blackout; "feeling seasick, having to weave as they walked", could not keep balance if stood still. |
| Bosphorus Bridge, Istambul[79] | Turkey | 2010 | |
| Bassac River Bridge[80] | Cambodia | 2010 | 456 people died in stampede after panic caused by swaying of bridge filled with over 7000 pedestrians trying to reach popular water festival |
| Westminster Bridge, London[34] | UK | 2010 | |
| Brooklyn Bridge[34] | USA | 2011 | |

The modelling and simulation process are illustrated schematically in Fig. 2. We have simulated three different variants of the pedestrian model. Model 1[18,31] is the simplest, based on linearising the inverted pendulum in the frontal plane for small angles. It assumes the sagittal-plane dynamics is independent of the lateral foot position and that foot transitions occur at regularly spaced prescribed times. At each transition the new lateral foot position is governed by a biophysically inspired control law[44] that enhances stability during horizontal ground motion. Model 2 is a new adaptation of Model 1, in which the timing of the foot placement alters as a kinematic consequence of the lateral bridge motion and foot placement. Finally, Model 3[20,45] assumes that the step timing is determined solely by the frontal-plane dynamics and that leg transition occurs each time the pedestrian CoM passes through a reference position defined as zero lateral displacement. A nonlinear feedback mechanism enables stable limit cycle motion in the absence of ground movement, and quasi-periodic motion on sinusoidally moving ground.

We choose parameters based on the set of controlled experiments on the London Millennium Bridge prior to reopening[22]. Up to $N = 275$ pedestrians were added individually at equally spaced time intervals $T_{add}$. We performed our simulations for two different choices of pedestrian addition times $T_{add} = 20$ s and $T_{add} = 10$ s. These choices are consistent with the incremental pedestrian loading tests on the London Millennium Bridge[22] and simulations conducted by Ingólfsson et al.[46] in which pedestrians were added at average intervals of 7 and 12 s, respectively.

The pedestrian parameters are drawn from distributions (see Table 3) and multiple simulations are run for different bridge and mean pedestrian frequencies. The number of pedestrians at which the vibration amplitude begins to increase rapidly is noted for each simulation. Representative results for $T_{add} = 20$ s are depicted in Fig. 3, with further results in the Supplementary Information (see Supplementary Fig. 1 for faster pedestrian addition time $T_{add} = 10$ s and Supplementary Fig. 2 for the worst-case scenario of complete resonance).

For each simulation, we numerically validate our general expression (1) for the total effective damping $c_T$ by calculating

$\sigma_1^{(i)}, \sigma_2^{(i)},$ and $\sigma_3^{(i)}$ for each pedestrian $i$ via (9)–(11). We also compute the Kuramoto order parameter[10] $r$, defined using

$$re^{i\psi} = \frac{1}{N}\sum_{i=1}^{N}\langle e^{i\varphi_i}\rangle, \qquad (3)$$

where $\varphi_i$ is the numerically calculated phase of the $i$th pedestrian's CoM or CoP (the distinction is made in Fig. 3), $\psi$ is the average phase, and $\langle \cdot \rangle$ denotes time average. Note that $r = 1$ implies complete synchrony, and $r = 0$ implies uncorrelated motion.

The simulations in Fig. 3 show how the onset of large amplitude bridge motion coincides with when the computed $c_T$ becomes negative, at $N = N_{crit}$. For Model 1, in which there is no adjustment to the gait frequency, the bridge's vibration amplitude grows unrealistically without bounds. In contrast, for Model 3, the onset of moderate amplitude motion starts a process of increased coherence (or phase pulling[15]) between the pedestrians' and bridge motion. The order parameter and inset sample solution traces indicate that increased synchrony then occurs between each pedestrian and the bridge. The amplitude of bridge vibrations then saturates. Model 2, which is a more realistic version of Model 1 for higher than moderate amplitude of bridge motion, shows similar amplitude saturation and coherence after instability occurs. Further simulations of Models 2 and 3 for different frequency parameters show that instability is at approximately $N = N_{crit}$ defined by (2), leading to a varying amount of synchrony as the amplitude grows. Thus, the negative-damping criterion can be understood as the cause of instability in all cases. Also, the varying degrees of synchrony are a consequence, not the cause, of the instability.

Note that the previous analysis[20] of the London Millennium Bridge instability based on Model 3 predicted the critical crowd size but, with some caveats, supported the synchronisation hypothesis. However, this analysis was performed for fixed crowd sizes such that a fixed number of pedestrians were placed on the bridge and the system was integrated for a sufficiently long time. Then, the crowd size was increased, and the simulations were repeated again. The key difference between these previous results[20] and our paper is that despite the strong propensity of

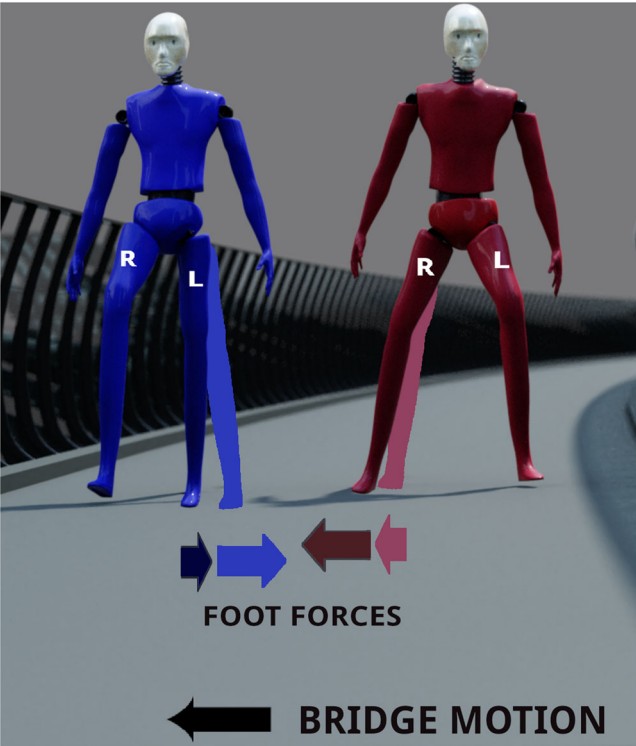

**Fig. 1 Explaining the fundamental mechanism underlying the negative damping owing to coefficient $\sigma_1$.** The figure contrasts the force transmitted to the bridge by two identical pedestrians who, when they simultaneously place their stance foot on the bridge (at the light blue and light red positions in an absolute co-ordinate frame), have equal and opposite gaits. As they place their feet, the lateral component of the foot force from each pedestrian is equal and opposite, so there is no net lateral force on the bridge. Suppose that during a time increment $\Delta t$ the bridge moves to the left, so that the blue figure's leg decreases its angle to the vertical within the frontal plane, whereas the red figure's leg angle increases. Thus, during this bridge motion, the magnitude of the lateral component of the red figure's lateral foot force increases whereas that of the blue figure decreases. Thus there is, on average, a change in resultant force in the direction of the bridge's motion. Nevertheless, there can be large variations depending on a pedestrian's foot placement strategy (see Figs. 5 and 6).

Model 3 for synchronisation, our results demonstrate that bridge instability occurs prior to the onset of crowd synchrony when pedestrians are added sequentially and the crowd size gradually increases in time, as in the controlled experiment on the London Millennium Bridge[22]. The previous work[17] also studied the London Millennium Bridge instability for fixed crowd sizes. In particular, this work used an energy-optimised pedestrian model with a linear feedback controller to demonstrate that heterogeneous pedestrians incapable of synchronising even at large crowd sizes can shake the bridge without synchronisation[17]. This effect was also reported in an earlier paper by Baker[28] and described for Model 1 in Macdonald[18]. Remarkably, our results indicate that pedestrians with a weak (Model 2) or strong (Model 3) propensity for synchronisation can first initiate the bridge vibrations at a critical crowd size and then become synchronised at larger crowd sizes when added sequentially (also see the extreme case of identical pedestrians in Supplementary Fig. 2 in the Supplementary Information).

**Frequency dependence.** A natural question is to seek to understand how the negative-damping coefficient depends on bridge

and mean pedestrian stride frequencies $\Omega$ and $\overline{\omega}$, and whether it can be enhanced or suppressed by resonance effects. Figure 4 shows the results of many ensemble runs. For each model we show in an upper plot the computed value of $\overline{\sigma}$ as a function of the ratio $\Omega/\overline{\omega}$ of bridge to average pedestrian frequency.

Note that Models 1 and 2 are effectively identical for small amplitude bridge motion. For Model 1, McRobie[47] derived an exact analytic expression for $\overline{\sigma}$ (shown as the green curve in the top panel of Fig. 4a). At the resonance condition where $\overline{\omega} = \Omega$ (represented by the yellow dot), the theory[47] predicts a large range of $\overline{\sigma}$-values, depending on the relative phase between the bridge and pedestrian. The hypotheses behind our general calculation of $\overline{\sigma}$ fail precisely at this resonance (see the Supplementary Information). The simulation results for Model 2 (represented by the blue dots), show features of large negative values of $\overline{\sigma}$ just below $\Omega/\overline{\omega} = 1$ and large positive values slightly above. These are believed to be due to the adaptation of the step timing, in response to perturbations from bridge motion, giving similar effects as previously found numerically for an inverted pendulum walking on a vertically oscillating structure[48] and experimentally for subjects walking on a laterally oscillating treadmill[15].

Also observe the paucity of data in certain regions of the lower panel of Fig. 4b and the apparent bi-modality of the data. This is because, for Model 3, limit cycle pedestrian motion is an emergent property of the simulations, rather than essentially an input parameter as it is for Models 1 and 2. Also note this model is liable to hysteresis between limit cycles of different period[45].

For all three models, we find the average value of $\overline{\sigma}$ to be mostly a function of the frequency ratio, being only a weak function of the pedestrian or bridge frequencies independently. Using this value in Eq. (2) gives the predicted critical number $N_{\rm crit}$ of pedestrians required to trigger an instability. The lower plots indicate the success of this prediction, by comparing it with the value of $N$ at which the vibration amplitude begins to increase rapidly in the simulations.

Also note the large spread of the model outputs for both $\overline{\sigma}$ and $N_{\rm crit}$, especially for Model 2. Our theoretical calculations only consider the long term averages of the effective damping coefficient $c_T$. This is only part of the story, because true walking behaviour is transient and involves changes to the trajectory of the walker's CoM and the foot placement strategy. On stationary ground, a walker's CoM will oscillate laterally with a dominant component at half the footfall frequency. Without changing the footfall frequency, the platform motion introduces a second frequency inducing the walker to adopt a two-frequency quasiperiodic pattern of footfall placement (Fig. 5). Depending on the phase of this quasiperiodic pattern, we have found that pedestrians can show large deviations from the long term average (see next section).

Nevertheless, for all three models, note that $N_{\rm crit}$ is minimised not when there is a frequency match between the pedestrian and bridge frequencies, $\Omega/\overline{\omega} = 1$, but when the pedestrian frequency is less than the bridge frequency, $\Omega/\overline{\omega} \approx 1.3$ for Models 1 and 2 and $\Omega/\overline{\omega} \approx 1.1$ for Model 3. Notice the red 5th percentile curves in Fig. 4 (top row) that indicate that negative damping can be observed at any frequency in the considered range of frequency ratios. Note too that there are some frequency ratios for which $\overline{\sigma}$ is positive. If pedestrians walked at those frequencies, then their motion would enhance that bridge mode's stability rather than reduce it.

An explanation of this frequency dependence can be summarised as being a question of timing. The argument in the caption of Fig. 1 implicitly assumes that the bridge is moving in a single direction during each step and that the bridge and pedestrian stride frequencies are similar. Particular tunings of this frequency ratio can in fact lead to a reversal of the effect in Fig. 1.

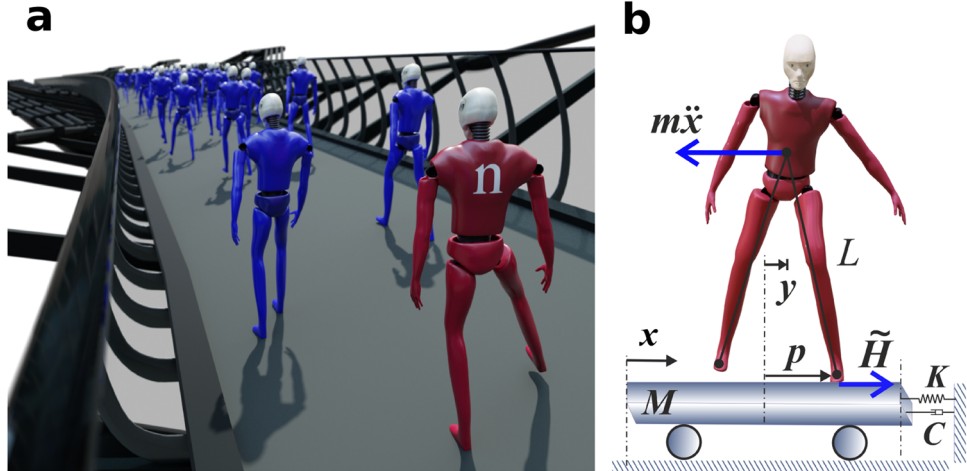

**Fig. 2 Outline of the mathematical model of pedestrian-induced lateral instability. a** Simulations are run for a coupled bridge-pedestrians system with pedestrians added sequentially at fixed time increments $T_{add}$ apart. The addition of the $n$th pedestrian ($n = N_{crit}$) causes the overall damping coefficient to become negative hence the amplitude of motion to increase rather than diminish. **b** Inverted pendulum model of bridge mode and pedestrian lateral motion. Here, $y$ is the lateral position of the pedestrian's centre of mass (CoM), while $p$ defines the lateral position of the centre of pressure (CoP) of the foot, both relative to the bridge. $L$ is the equivalent inverted pendulum length and $m$ is the pedestrian mass. The displacement $x$ of the bridge in a lateral vibration mode is represented by an equivalent platform with mass $M$, spring constant $K$ and damping coefficient $C$. $\tilde{H}$ is the lateral component of the pedestrian's foot force on the bridge deck. In return, the bridge motion causes an inertia force $-m\ddot{x}$ on the pedestrian's centre of mass. The pedestrians are depicted as "crash test" dummies with flexible hips; however, the actual inverted pendulum model is simpler, with pendulum-like legs connecting to the CoM.

**Table 3 Default parameter values used in the simulations. Here, S.D. is the standard deviation of parameter mismatch among pedestrians, which follows a normal distribution in all cases.**

| Parameter | Meaning | Units | Default value | Mismatch S.D. | Source |
|---|---|---|---|---|---|
| $a$ | Auxiliary | m | 0.047 | 0 | Ref. [20,45] |
| $b_{min}$ | Margin of stability | m | 0.0157 | 0.002 | Ref. [18] |
| $C$ | Bridge damping | Ns/m | 29,251 | | |
| $g$ | Acceleration of gravity | m/s$^{-2}$ | 9.81 | | |
| $L$ | Effective leg length | m | 1.17 | 0.092 | Ref. [31] |
| $m$ | Pedestrian mass | kg | 76.9 | 10 | Ref. [31] |
| $M$ | Bridge mass | kg | 113,000 | | Ref. [11] |
| $p_c$ | Auxiliary | m | 0.063 | 0 | Ref. [20,45] |
| $T_{add}$ | Pedestrian addition time | s | 20 | | |
| $X_0$ | Unperturbed half step length | m | 0.36 | | Ref. [42,81] |
| $Y_0$ | Unperturbed half step width | m | 0.047 | | Ref. [18] |
| $\lambda$ | Damping due to walking | s/m$^2$ | 23.25 | 0 | |
| $\omega$ | Unperturbed angular Stride frequency | rad/s | 5.655 | 0.1 | Ref. [20,42] |
| $\Omega$ | Angular bridge Natural frequency | rad/s | 6.503 | | Ref. [11,12] |

Nevertheless, over the frequency range considered, both the size of the regions of pedestrian-induced negative damping and its average value greatly outweigh that of positive damping.

**The role of foot placement strategies**. Figure 1 explains how bridge motion breaks the symmetry of the loading applied by mirror-imaged walkers such that long-term averages need not equal zero. That is only part of the explanation, as it does not consider the motion of the walkers' centres of mass nor the various foot placement strategies that may be adopted to maintain balance. In principle, the foot placement as defined by Hof et al.[44] is dependent on the lateral velocity of the pedestrian's centre of mass. However, uncertainty remains as to whether the velocity should be defined in reference to the oscillating bridge (relative velocity) or a stationary point against which the bridge is moving

(absolute velocity). Therefore, Fig. 5 shows results from Model 1 for both of these conditions.

The corresponding forces applied to the bridge in these three cases are also shown in Fig. 5. Since bridge motions are small, the forces are similar in all three cases. By taking the difference in forces, Fig. 6 highlights the small change in the applied forces that are the result of the bridge motion, and correlates these with bridge velocity. The walker adopting the relative velocity control law creates forces which are negatively correlated with bridge velocity, leading to a positive damping effect. By contrast, the additional forces generated by the walker adopting the absolute velocity balance law are positively correlated with the bridge velocity, leading to the negative damping effect which feeds energy into the bridge.

In summary, the bridge motions cause the walkers to adjust their foot placements which induces small quasiperiodic forces

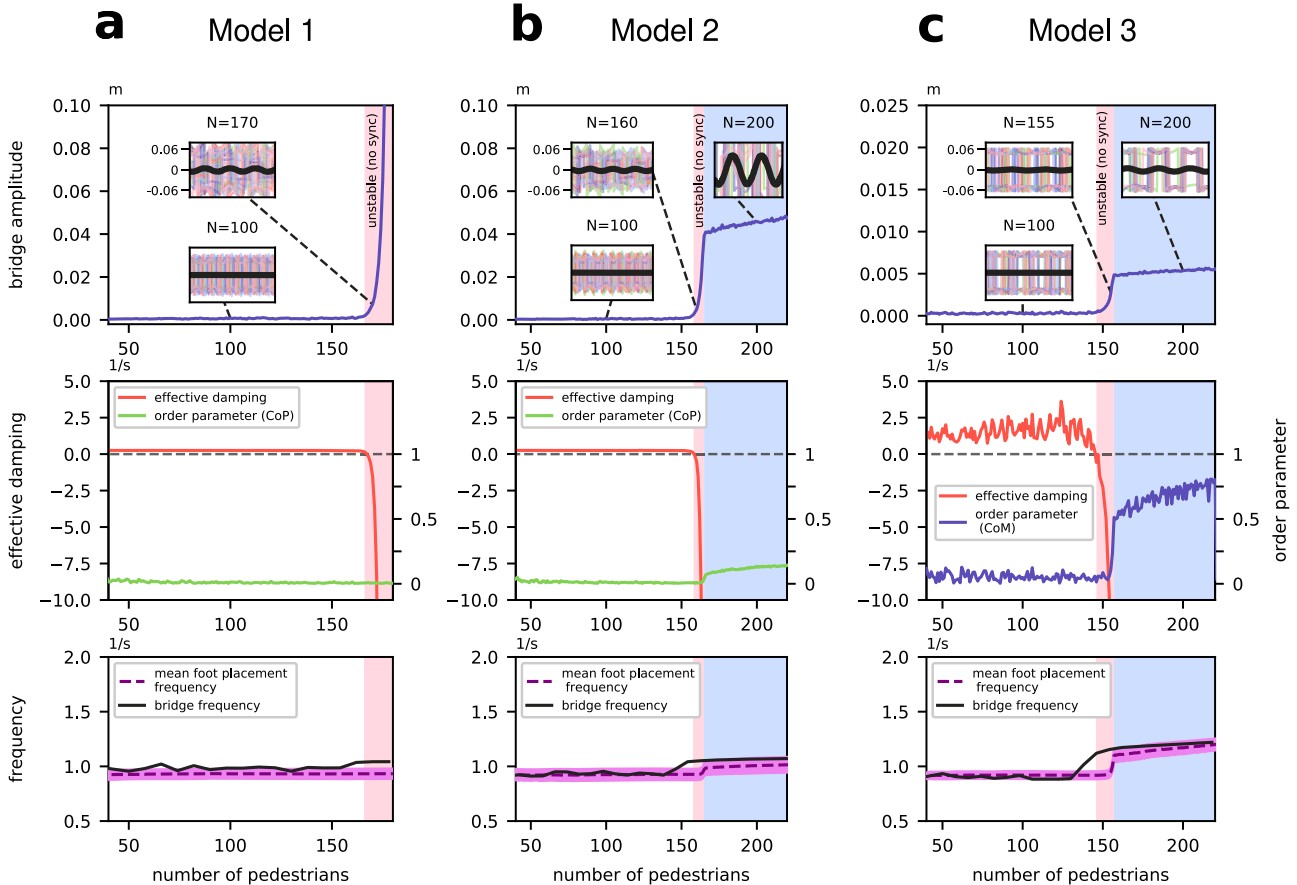

**Fig. 3 Example simulations showing the nature of the bridge instability for each of our three models.** See Methods for model details and parameter values. (Top row): Bridge vibration amplitude as a function of number of pedestrians $N$. The left-hand boundary of the pink shaded portion indicates the value $N_{\mathrm{crit}}$ where $c_T$ crosses zero, and the blue shaded portion is where a degree of synchrony is observed. Insets show illustrative bridge $x(t)$ (black) and a few representative pedestrian $y(t) - p(t)$ (coloured) oscillations over three cycles. (Middle row): Computation of the total bridge damping $c_T$ given by Eq. (1) and the Kuramoto order parameter $r$ given by Eq. (3) calculated for the phases of pedestrians' CoP (Models 1 and 2) and CoM (Model 3). (Bottom row): instantaneous computed bridge and pedestrian foot placement frequencies. **a** Simulations of Model 1 which cannot synchronize. **b** Simulations of Model 2 which permits weak synchronization. **c** Simulations of Model 3 with strong propensity for synchronization.

which have a component at the bridge frequency. Depending on the balance law adopted (and the frequency of bridge motion and other parameters), the phases of these additional forces can either add or extract energy to/from the bridge.

Experimental evidence is limited as to which balance law is more realistic for a walker on a moving platform, but the laboratory experiments augmented with Virtual Reality by Bocian et al.[15] provide some evidence for the absolute velocity control law. Walkers following either law could be present on the bridge. Also, the energy flows vary within different regimes of the quasiperiodic motions, such that the short-term effective damping may vary markedly from its theoretical long-term average value. Bridge designers should thus be aware that there could be dangerous instances of the negative damping effect at any bridge frequency.

This is the underlying cause of the instability of footbridges and does not entail walkers making any change to the frequency of their footsteps. Instead, gait widths are amplitude modulated, introducing complicated phase relationships between foot placements and bridge motions, many of which have the effect of negative damping and feed energy into the bridge.

As bridge amplitudes grow, adjustment of footfall timing is an additional possibility and this is included in Models 2 and 3. Potential outcomes include the now-classical Kuramoto transition to synchronisation, as well as phase pulling phenomena

where footfalls do not fully synchronise to the bridge motions, but spend proportionally longer at some relative phase offsets[15,34]. Walkers who synchronise or exhibit phase pulling can add differing amounts of energy to the bridge, depending how their footfall phases relate to that of the bridge velocity. Phase synchronisation can be triggered by bridge motions excited by the more fundamental mechanism of amplitude-modulated gait width, and this can lead to dangerous amplification of the bridge motions. It may also be noted that there exist parameter regimes where walkers synchronise at phases that lead to energy absorption or where they synchronise with a certain phase but the amplitude of the forcing does not grow indefinitely with the bridge amplitude, thereby limiting the bridge response. However, there is insufficient evidence for this to be relied upon in bridge design.

**Discussion**
In conclusion, the question of what caused the instability of the London Millennium Bridge on its opening day can be referred to as a debate in the literature between the negative damping and synchronisation hypotheses. The main contribution of this paper has been to show that the view that the instability of the London Millennium Bridge on its opening day was caused by a textbook example of synchronisation of coupled pedestrians is wildly inaccurate, at best misguided and if used to try to design

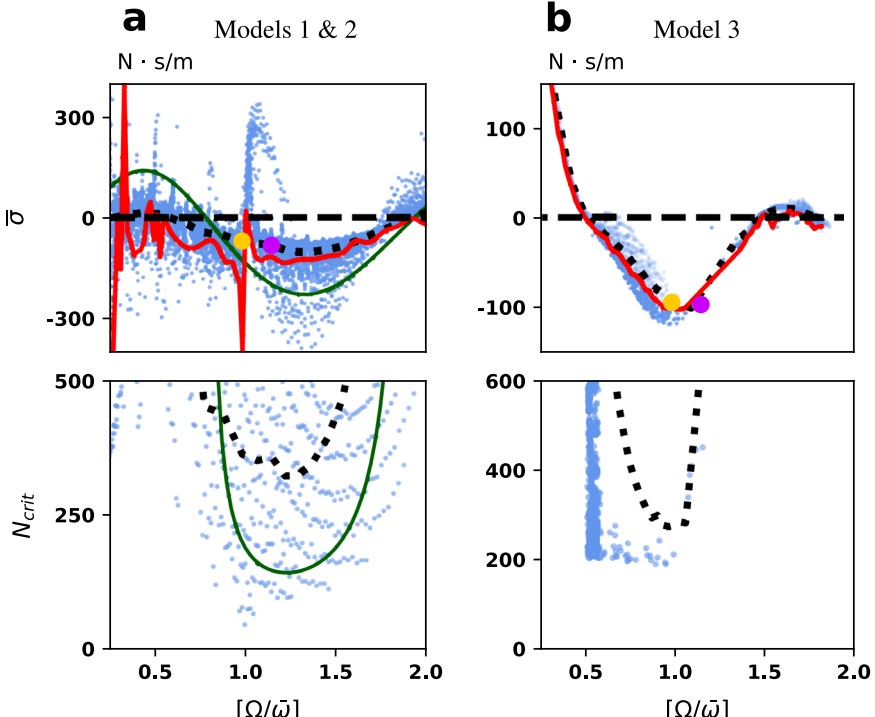

**Fig. 4 Average damping coefficient per pedestrian $\bar{\sigma}$ calculated via (35), given in the Supplement Information (top row) and the critical crowd size $N_{crit}$ (bottom row) as a function of numerically calculated bridge and pedestrian frequencies ratio [$\Omega/\bar{\omega}$].** Simulations of Models 1 and 2 (**a**) and 3 (**b**) indicate the range of frequency ratio [$\Omega/\bar{\omega}$] in which $\bar{\sigma}$ is negative so that a single pedestrian, on average, contributes to bridge instability. Each ratio of [$\Omega/\bar{\omega}$] corresponds to different combinations of $\Omega$ and $\bar{\omega}$ (blue dots). Black dotted lines indicate the average of $\bar{\sigma}$ and $N_{crit}$ for a given ratio. The red curve indicates the 5th percentile of the $\bar{\sigma}$ distribution. The green curve is the analytical expression (36) for $\bar{\sigma}$ (top plot) and analytical estimate (37) for $N_{crit}$ (bottom plot), given in the Supplementary Information and calculated for Model 1 with identical pedestrians with fixed $\omega = 5.655$ rad/s and S.D. = 0. The magenta dot corresponds to the initial ratio [$\Omega/\bar{\omega}$] used in Fig. 3, the yellow dot corresponds to $\Omega/\bar{\omega} = 1$. See the Supplementary Information for the details of the calculations.

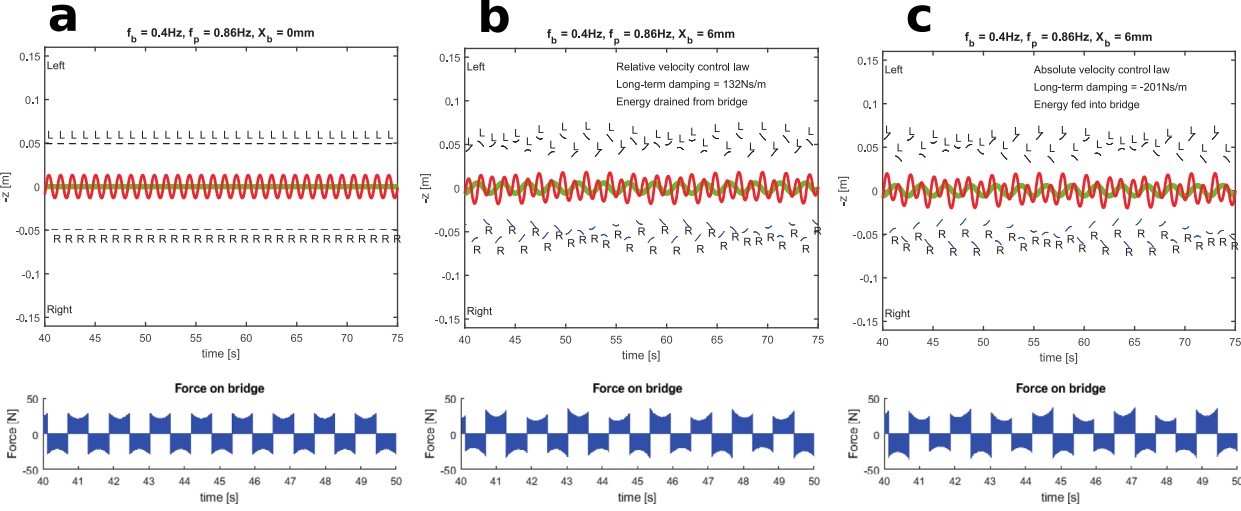

**Fig. 5 Upper panels show foot placement patterns (short black lines left foot, short blue lines right foot) for Model 1.** Panel **a** is for a stationary platform, while panels **b** and **c** are for a bridge oscillating at 6 mm amplitude at 0.4 Hz, with walkers adopting Hof et al.'s[44] balance laws based on relative and absolute velocity, respectively. The bridge motions induce quasiperiodic placement patterns. The walker's centre of mass and the bridge displacements are shown in red and green, respectively. The lower panels show the corresponding forces applied to the bridge. Walker parameters: $m = 74.4$ kg, $f_{walk} = 0.86$ Hz, $L = 1.2$ m, $b = 15.7$ mm.

mitigation strategies in terms of frequency avoidance, potentially dangerous.

Indeed, even when much is known about the physical properties of a bridge, knowledge of the crowd behaviour is necessarily subject to large uncertainties, both aleatoric and epistemic. For example, not only will there be a distribution of foot-placement control laws amongst the individuals in any crowd, but that distribution is not known. Despite this inevitable uncertainty, it is still possible to make quantitative statements. A specific point is that bridges with low natural frequencies (close to say 0.4 Hz,

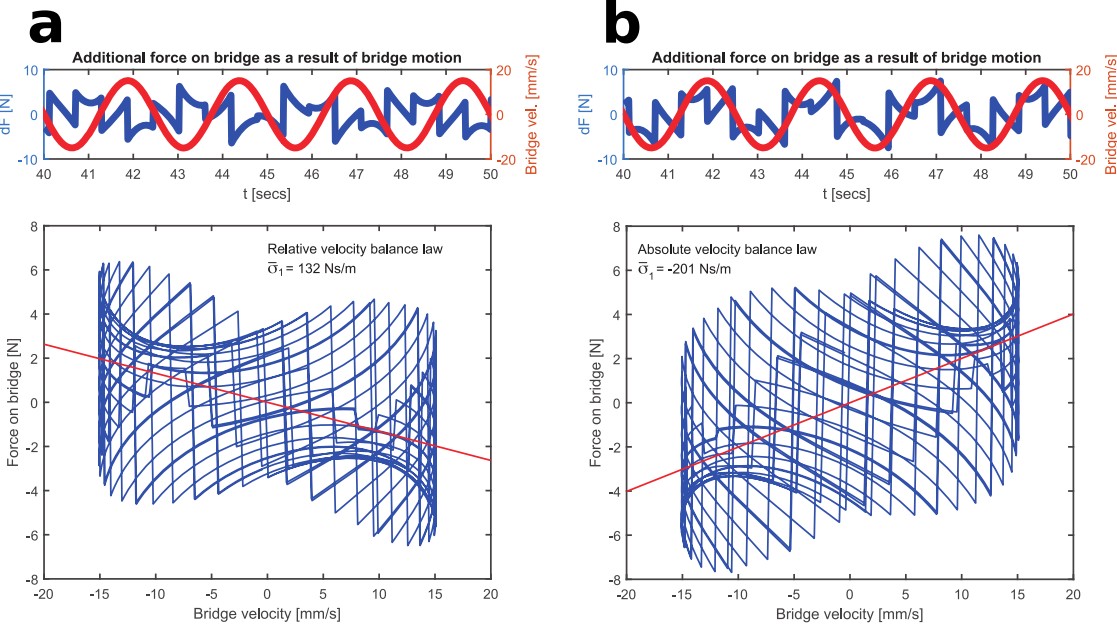

**Fig. 6 Upper panels: the change in forces that are the result of the bridge motions for the walkers of Fig. 5.** The bridge velocity is shown in red. Lower panels: the correlation between the bridge velocity and the induced forces. The red lines indicate the average effective damping coefficient $\bar{\sigma}$. Panel **a** corresponds to panel **b** in Fig. 5; panel **b** corresponds to panel **c** in Fig. 5.

which is much lower than the dominant lateral excitation frequency, circa 1 Hz) would not be expected to be excited by a crowd according to the main synchronisation hypothesis, since it is arguably unlikely that an individual would slow the cadence of their footfalls by a factor of 2.5 to synchronise. If that were accepted, bridge designers could thus argue that no precautions need be taken for low frequency bridges against the possibility of lateral excitation phenomena, whereas the models analysed here show that this is far from the case. Preventative measures such as tuned mass dampers are expensive, and there are incentives for arguing that they are not necessary; our work shows that this would be a dangerous path to take. This paper's demonstration of the alternative paradigm shows that the frequency range of concern is much wider than implied by some earlier theories, and the inherent uncertainties make this frequency range wider yet. Note how our scatter plots of Fig. 4 provide quantitative illustrations of this. Calibration of the models and inclusion of further features such as mode shapes and possible pedestrian-to-pedestrian interactions in dense crowds may lead to improved guidelines for bridge design. In particular, crowd congestion can cause footfall frequencies to enter into bands that are more likely to trigger instability[43], or human-to-human interactions may affect footstep timing. Our asymptotic formulae are well suited for addressing these research questions as the contribution of social force pedestrian dynamics[43] in promoting or damping instability can be explicitly evaluated via integral quantity $\sigma_3$. These calculations are a subject of future work.

A key scientific conclusion of this paper has been to argue that negative damping due to pedestrians' attempts to maintain balance is in most cases likely to be the essential cause of lateral bridge instability. Moreover, any synchronisation is typically a consequence, rather than a cause, of the instability. Indeed, in our simulations we observed that increased synchrony, or more accurately increased coherence, among pedestrians' foot placements is part of a secondary nonlinear adjustment to the amplitude of vibration after the instability has been initiated. This secondary effect in most cases causes saturation of the vibration amplitude but can, in extreme cases, further exacerbate the instability.

These findings have been achieved through asymptotic analysis applicable to a wide class of foot force models, and are demonstrated using three specific models, one which cannot synchronise, one that includes adaptation that permits synchronisation, and one which is highly prone to synchronisation. Moreover, we have conducted a comprehensive review of the literature on real bridges that have experienced large amplitude lateral pedestrian-induced vibrations. It is clear from this review that any direct evidence of synchronisation is at best scant. In contrast, our theory is fully consistent with all known observations.

Nevertheless, that the problem is subtle is something that we have tried to emphasize. Increased coherence among pedestrian footsteps can occur, especially if pedestrians happen to be walking close to a natural frequency of the bridge. Indeed, previous papers that have purported to show synchronisation as being causal for the bridge instability have focused exclusively on that case[17,49]. But even in those cases where there is significant coherence in pedestrian behaviour as bridge amplitude grows, our simulations suggest that negative damping can still be regarded as the trigger of the instability. See, for example, the results of Model 3 in Fig. 3, and the even more extreme case in Supplementary Fig. 2 in the Supplementary Information, where negative damping precedes the onset of bridge amplitude growth and subsequent synchronisation, upon adding pedestrians sequentially.

Our findings should enable bridge designers and other structural engineers to develop more accurate design criteria to avoid human-induced instability of a wide range of structures. Unfortunately, our results show there is no magic formula for certain lateral frequencies to avoid when designing a bridge. The negative damping-induced instabilities are not restricted to cases where lateral bridge modes are close to resonance with pedestrian walking frequencies. In truth, there is no substitute to ensuring that there is sufficient lateral damping in the bridge design. Nevertheless, our asymptotic formulae can at the very least provide estimates for the level of damping required, given the expected number of pedestrians using the bridge.

Note that a negative-damping instability can be viewed mathematically as an example of a Hopf bifurcation,

characterised by a complex conjugate pair of eigenvalues of the bridge dynamics crossing the imaginary axis[50]. An analogous instability is well known in fluid-structure interaction, where it is called flutter.

At a more general level, our results point to an alternative kind of emergent behaviour among autonomous agents. The usual theory of synchronisation distinguishes between cases where there is a master conductor that all other agents follow, and where synchrony emerges spontaneously without a leader. We have uncovered a third possibility, that there is an underlying, albeit nascent, collective frequency that does not become excited until the individual agents are sufficiently active. Each agent need not synchronise to the collective frequency, nor to another agent. Each agent simply needs to display some positive feedback effect. An intuitive, yet erroneous, argument might suggest that in the absence of coherence, the feedback from all the agents would, on average, cancel each other out. But this is not how positive feedback works, it creates a bias that can lead to negative damping.

This kind of emergent instability may actually be more prevalent in nature and society than previously thought. For example, both in the mammalian[51] and insect[52] hearing systems, single-frequency instability of an active system can occur due to beating of tiny incoherent neuro-mechanical oscillators. In the mammalian system, for example, the active neuro-mechanical oscillators in question are the so-called outer hair cells. Small heterogeneities in the properties of the tuning can cause a Hopf bifurcation to occur leading to so-called otoacoustic emissions to be radiated out from the ear canal in the absence of any stimulus.

Another example of this kind of instability may be in how macroeconomic and financial systems tend to develop characteristic cycles[53] without there being obvious causal synchrony at the microeconomic level. Such propensity of economic systems made up of many uncorrelated microeconomic components with different intrinsic properties to give rise to macro-scale boom and bust cycles, has been modelled mathematically using a so-called Goodwin oscillator which can be represented mechanically as a so-called Phillips machine[54]. Here, the global economy is likened to a continuum of micro-scale fluid particles. At the macro-scale, the system goes unstable due to an analogue of fluid-structure interaction flutter, which in this case is actually a kind of non-smooth Hopf bifurcation.

## Methods

**Mathematical models**. When considering possible mechanisms by which pedestrians could be prompted to generate synchronised loading onto the bridge, it seems that the pedestrian–structure rather than pedestrian-pedestrian interaction is dominant[24]. Visual and auditory stimuli on their own do not lead to significant levels of spontaneous synchronisation within a group of pedestrians walking on stationary ground[55]. From the perspective of functional human gait, synchronisation cannot be considered as one of the fundamental qualities of locomotion, unlike stability, which is critical[56]. Therefore, the primary objective of pedestrians walking on vibrating ground is to remain balanced. In the case when medio-lateral gait stability is challenged, this is mainly achieved by adapting the step width, and a large body of evidence already exists supporting this notion (e.g.,[57,58]). In line with this evidence, our mathematical model simply supposes that any possible movement coordination between pedestrians is due solely to sensory stimuli from the moving ground and the associated mechanical feedback.

The displacement of the lateral bridge mode $x(t)$ is assumed to be governed by a simple second-order equation of motion

$$M\ddot{x} + C\dot{x} + Kx = \sum_{i=1}^{N}\tilde{H}^{(i)}(x, y^{(i)}), \quad (4)$$

where $M$, $C$, and $K$ are the mass, damping and stiffness coefficients, respectively, of the bridge mode and $y^{(i)}(t)$ is the lateral displacement of the centre of mass of the $i$th pedestrian, relative to the bridge. The forcing term $\tilde{H}^{(i)}$ is the lateral component of the $i$th pedestrian's foot force on the bridge deck.

A number of models of varying complexity may be used to capture the motion of a pedestrian in response to ground movement[56–58]. Here, we seek only to model the lateral component of each pedestrian's foot force on the bridge. To do this, we

make the simple assumption that the lateral component of the centre of mass of a pedestrian of mass $m$ obeys an equation of the form

$$m\ddot{y}^{(i)} + m\ddot{x} = -\tilde{H}^{(i)}(x, y^{(i)}), \quad i = 1, \dots N. \quad (5)$$

In general, $\tilde{H}^{(i)}$ is a function of exogenous variables associated with the pedestrian's gait, particularly the lateral motion, and will typically be a piecewise-smooth function with abrupt changes at foot transitions. Specifically, we assume that foot transitions occur at a sequence of times $\{t_s^{(i)}\}$, $s = 1, 2, 3, \dots$, where $t_{s+1}^{(i)} > t_s^{(i)}$ for all $s$. By definition the angular pedestrian stride frequency is $[\omega_i] = 2\pi/[(t_{s+2}^{(i)} - t_s^{(i)})]$, where $[\cdot]$ denotes possible adjustment due to bridge motion. For definiteness, we assume even $s$ corresponds to touchdown of the right foot and odd $s$ to touchdown of the left.

Our analysis of negative damping is applicable to any model that can be written in the form (4) and (5). It is helpful to scale parameters and introduce dimensionless parameters $\varepsilon$ and $\zeta$ measuring mass and damping ratios respectively

$$H^{(i)} = \tilde{H}^{(i)}/m, \qquad \Omega = \sqrt{K/M}, \qquad \varepsilon^2 = m/M, \qquad \zeta = \frac{C}{2M\Omega\varepsilon}. \quad (6)$$

Then the equations of motion can be written in the form

$$\ddot{x} + 2\varepsilon\Omega\zeta\dot{x} + \Omega^2 x = \varepsilon^2 \sum_{i=1}^{N} H^{(i)}, \qquad \ddot{y}^{(i)} + H^{(i)} = -\ddot{x}, \quad i = 1, \dots, N. \quad (7)$$

Note the modelling choice that the bridge's natural damping in (7) is assumed to be $\mathcal{O}(\varepsilon)$. This is consistent with values of bridge damping and numbers of pedestrians $N = \mathcal{O}(\varepsilon^{-1})$ required to trigger instability observed in practice (see the Supplementary Information).

Treating $\varepsilon$ as a small parameter, a lengthy, but straightforward multiple-scale asymptotic expansion (see subsection Asymptotic derivation of negative damping criterion) can be used to evaluate the total bridge damping as the natural damping plus three additional terms:

$$c_T = 2\varepsilon\zeta\Omega + \varepsilon\nu(\overline{\sigma}_1 + \overline{\sigma}_2 + \overline{\sigma}_3) = 2\varepsilon\zeta\Omega + N\varepsilon^2 \sum_{i=1}^{N}(\sigma_1^{(i)} + \sigma_2^{(i)} + \sigma_3^{(i)}), \quad (8)$$

with

$$\sigma_1^{(i)} = \frac{1}{T_i}\int_0^{T_i}\frac{\partial H^{(i)}}{\partial\dot{x}}dt, \quad (9)$$

$$\sigma_2^{(i)} = \frac{1}{T_i\Omega}\left(\overline{y}_s^{(i)}\int_0^{T_i}\frac{\partial H^{(i)}}{\partial y}dt + \Omega\overline{y}_c^{(i)}\int_0^{T_i}\frac{\partial H^{(i)}}{\partial\dot{y}}dt\right), \quad (10)$$

$$\sigma_3^{(3)} = \frac{1}{T_i\Omega}\left(\overline{z}_s^{(i)}\int_0^{T_i}\frac{\partial H^{(i)}}{\partial z}dt + \Omega\overline{z}_c^{(i)}\int_0^{T_i}\frac{\partial H^{(i)}}{\partial\dot{z}}dt\right). \quad (11)$$

Here, a subscript $c$ means component in phase with the bridge instantaneous displacement ($c$ stands for cosine) and $s$ means component in anti-phase with the bridge velocity ($s$ stands for sine). Also an overline means time average over many steps. Furthermore, $z(t)$ is the perturbation, due to the lateral motion, of the pedestrian's forward position relative to a constant forward speed. Because each function $H^{(i)}$ is in general nonsmooth, partial derivatives should be interpreted in the distributional sense (see the Supplementary Information).

The particular pedestrian models we use in our simulations are distinguished only by their choice of the foot force function $H^{(i)}$, which we assume to take an identical form for each pedestrian, but to have parameters that can vary between pedestrians.

*Model 1: Linearised inverted pendulum with step width control.* This model was developed by Macdonald, Bocian, and Burn[18,31] and was shown to exhibit similar features to those observed in four independent experimental studies[15,16,32,33,59]. Here

$$H^{(i)}(t) = \frac{g}{L}(p^{(i)}(t_s) - y^{(i)}), \quad (12)$$

with $g$ being gravitational acceleration and $L$ effective leg length, and $p^{(i)}(t_s)$ is the lateral centre of pressure of the foot placed at time $t_s$. At the beginning of each step, $p^{(i)}(t_s)$ is adjusted according to the self-balancing control law determined theoretically and experimentally by Hof et al.[44,60]:

$$p^{(i)}(t_s) = y^{(i)}(t_s^-) + \sqrt{\frac{L}{g}}(\dot{y}^{(i)}(t_s^-) + \kappa_1\dot{x}_0(t_s^-)) + (-1)^s b_{\min}, \quad (13)$$

where $t_s^-$ is the time immediately before foot transition, and $b_{\min} > 0$ is the margin of stability, proportional to the natural gait width in the absence of any bridge motion. Whether the foot placement control law depends on the velocity $\dot{y}^{(i)}$ of the walker's centre of mass relative to the bridge motion $\dot{x}_0$ or the absolute velocity $\dot{y}^{(i)} + \dot{x}_0$ is set by the parameter $\kappa_1$, with $\kappa_1 = 0$ or $\kappa_1 = 1$ corresponding to relative or absolute velocity control laws, respectively. In this model, the walking frequency that defines the switching times $t_s$ is given by an external clock and is not adjusted due to bridge motion. Thus, each $\omega_i$ remains constant throughout the simulation.

*Model 2: Model 1 with step-timing adaptation.* We introduce adaptation to the step time $t_s$ due to the geometric nonlinearity associated with the adjustment to the lateral gait width. Consider a rigid, three-dimensional inverted pendulum of length $L = \sqrt{X^2 + Y^2 + Z^2}$, where $X$, $Y$, and $Z$ represent, respectively, displacements of the centre of mass, relative to the centre of pressure (CoP) of the stance foot, in longitudinal, transverse, and vertical pedestrian-centred coordinates. Suppose $X(t_s^-) = X_0 + \Delta X$, where $(X_0, Y_0, Z_0)$ is the position of the centre of mass at touchdown of the next foot for unperturbed steady state walking. Assume that, with perturbations from bridge motion, foot transition still occurs when $Z = Z_0$, then

$$X(t_s^-)^2 + Y(t_s^-)^2 = X_0^2 + Y_0^2,$$

where $Y(t_s^-) = y^{(i)}(t_s^-) - p^{(i)}(t_{s-1})$ is the transverse position of the centre of mass, relative to the CoP, just before touchdown, with $p^{(i)}(t_{s-1})$ from the previous foot transition from (13). Hence, in the limit of small $\Delta X$, we can write

$$\Delta X = \frac{1}{2X_0}(Y_0^2 - Y(t_s^-)^2). \quad (14)$$

Introducing the mean forward velocity

$$\chi = \frac{2X_0}{\pi/\omega_i} = \frac{2}{\pi}X_0\omega_i, \quad (15)$$

the perturbation to the timing of the next step is approximately $\Delta t = \Delta X/\chi$, hence the time of the next step is given by

$$t_s = t_{s-1} + \frac{\pi}{\omega_i} + \frac{\Delta X}{\chi} = t_{s-1} + \frac{\pi}{\omega_i}\left[1 + \frac{Y_0^2 - \{y^{(i)}(t_s^-) - p^{(i)}(t_{s-1})\}^2}{4X_0^2}\right].$$

Supplementary Movie 1 displays a pedestrian walking according to Model 2 subject to an imposed sinusoidal bridge motion with an amplitude of 1 cm and a frequency of 1.039 Hz close to that of the London Millennium Bridge. In Supplementary Movie 1, the motions of the CoM and CoP of the two-legged inverted pendulum and its 3D humanoid avatar are governed by numerically calculated $y(t)$ and $p(t_s)$ from Model 2. Note that the legs of the 3D humanoid avatar do not connect at the body centre of mass, but have a finite hip width. This hip width is not in the mathematical model. Only the CoM and CoP are modelled, with a rigid—though not necessarily direct straight—connection between them. The legs in the animation, though not drawn on a direct straight line between the CoM and CoP, connecting them rigidly, and the CoM and CoP lateral positions are exactly as found from the model.

*Model 3: Rocking inverted pendulum.* We have also implemented the autonomous walking model proposed and studied by Belykh et al.[20,45] that displays stable limit cycle motion without the need for any control. Here

$$H = \lambda\left[\dot{y}^2 + \frac{g}{L}\{a^2 - (y - p_c sgn(y))^2\}\right]\dot{y} - \frac{g}{L}(y - p_c sgn(y)), \quad (16)$$

where, in contrast to Models 1 and 2, the lateral position of the CoP of the foot $p$ is a fixed margin, denoted by constant $p_c$. Here, $\lambda$ is a damping parameter, $a$ is a parameter that controls the amplitude and the period of the limit cycle. In the absence of bridge motion, the amplitude and period of the limit cycle can be calculated explicitly.[45]

Unlike Models 1 and 2, the times at which the system with footforce (16) switches legs depends on the lateral motion of the centre of mass, rather than the forward walking speed. That is, leg transition occurs whenever $y$ crosses zero. Thus, the walking frequency adapts in the presence of bridge motion.

**Asymptotic derivation of damping criterion.** Our aim is to derive a general expression for the total bridge damping for a general model of the form (7), as a function of the number of pedestrians. Hence, we seek to find the number of pedestrians $N_{crit}$ required for instability.

The method we use is that of multiple scale asymptotic expansions. This is a standard technique within applied mathematics and can be used to estimate the amplitude of weakly nonlinear vibrations[61]. The basic idea is to find a balance between the bridge's natural damping and the ratio of a typical pedestrian mass and the modal mass of the bridge mode of vibration in question. Parameters are then rescaled according to a small parameter $\varepsilon$ that measures the size of these effects. Then, one is able to calculate the total adaptation $\bar{\sigma}$ to the bridge's effective damping from each pedestrian, averaged over many steps. Finally, one averages over an ensemble of pedestrians to find the critical number $N_{crit}$ that are necessary on average to reduce the effective damping to zero. We shall present an outline of the calculation here, with the details relegated to the Supplementary Information.

In this section all frequencies are assumed to be angular frequencies in units of radians per second. We shall discover that $N_{crit} = \mathcal{O}(\varepsilon^{-1})$, hence it will be convenient in what follows to write

$$N = \nu\varepsilon^{-1}, \quad \text{where} \quad \nu = \mathcal{O}(1). \quad (17)$$

We shall assume that the forward motion of the pedestrian's centre of mass can also be described by a single degree of freedom $z^{(i)}$. Thus the general dimensionless

model can be written in the form

$i$ th-pedestrian lateral motion: $\quad \ddot{y}^{(i)} + H^{(i)}(x, \dot{x}, y^{(i)}, \dot{y}^{(i)}, z^{(i)}, \dot{z}^{(i)}) = -\ddot{x}, \quad (18)$

$i$ th-pedestrian forward motion: $\quad \ddot{z}^{(i)} + G^{(i)}(y^{(i)}, \dot{y}^{(i)}, z^{(i)}, \dot{z}^{(i)}) = 0, \quad (19)$

single lateral bridge mode: $\quad \ddot{x} + \varepsilon 2\zeta\Omega\dot{x} + \Omega^2 x = \varepsilon^2 \sum_{i=1}^{N} H^{(i)}(x, \dot{x}, y^{(i)}, \dot{y}^{(i)}, z^{(i)}, \dot{z}^{(i)}). \quad (20)$

Here, $G^{(i)}$ is a general nonlinear function of its arguments and, like $H^{(i)}$, is typically nonsmooth.

In the absence of bridge motion, we assume that the pedestrian dynamics

$$\ddot{y}^{(i)} + H^{(i)}(0, 0, y^{(i)}, \dot{y}^{(i)}, z^{(i)}, \dot{z}^{(i)}) = 0, \quad \ddot{z}^{(i)} + G^{(i)}(y^{(i)}, \dot{y}^{(i)}, z^{(i)}, \dot{z}^{(i)}) = 0$$

admits an asymptotically stable limit cycle with period $T_i = 2\pi/\omega_i$

$$y^{(i)} = y_0^{(i)}(t), \quad y_0^{(i)}(t) = y_0^{(i)}(t + T_i), \quad z_0^{(i)} = \chi t + z_0^{(i)}(t + T_i),$$

where $y_0$ and $z_0$ are periodic functions of time, and $\chi$ is the average forward velocity of the pedestrian's centre of mass. Moreover, we suppose that

$$H^{(i)}(0, 0, y_0^{(i)}, \dot{y}_0^{(i)}, z_0^{(i)}, \dot{z}_0^{(i)}) = h_0^{(i)}(t), \quad \text{and} \quad G^{(i)}(y_0^{(i)}, \dot{y}_0^{(i)}, z_0^{(i)}, \dot{z}_0^{(i)}) = g_0^{(i)}(t)$$

are $T_i$-periodic functions.

We begin with a technical, detuning assumption that simplifies the analysis, namely that each pedestrian has an independent frequency $\omega_i$, and that there exists a constant $R > 0$ such that

$$\min_{i \neq j}|\omega_i - \omega_j| > R\varepsilon, \quad \min_i|\omega_i - \Omega| > R\varepsilon. \quad (21)$$

We look for a coupled solution to the system (18)–(20) as an asymptotic expansion in $\varepsilon \ll 1$ of the form

$$x = \varepsilon x_1(t) + \varepsilon^2 x_2(t) + \dots, \quad y^{(i)} = y_0^{(i)}(t) + \varepsilon y_1^{(i)}(t) + \dots, \quad z^{(i)} = \chi t + z_0^{(i)}(t) + \varepsilon z_1^{(i)}(t) + \dots. \quad (22)$$

Details of the computation of each term in this expansion are presented in the Supplementary Information. We then use the well-known method of multiple scales[61] under the assumption that the free vibration of the bridge can be written in the form

$$x_1(t) = X(\tau)\cos(\Omega t + \phi(\tau)),$$

where $\tau$ is a slow timescale which is affected by the motion of each pedestrian. We then consider the next-order perturbation $y_1(t)$ and $z_1(t)$ to the pedestrian motion and feed this back into the second-order equation for the bridge motion. The requirement that there should be no secular terms (proportional to $\sin(\Omega t)$ and $\cos(\Omega t)$) then gives a solvability condition for $X$ and $\phi$. The details of this process are given in the Supplementary Information.

We finally arrive at

$$\phi' = -\frac{\nu}{\Omega}(\hat{h}_x + \hat{\kappa}_y + \hat{\kappa}_z), \quad (23)$$

$$\frac{X'(\tau)}{X} = -2\zeta\Omega - \frac{\nu}{\Omega}(-\Omega\hat{h}_x + \hat{\sigma}_y + \hat{\sigma}_z), \quad (24)$$

where

$$\hat{\kappa}_p = \frac{1}{N}\sum_{i=1}^{N}\int_0^{T_i}(h_p^{(i)}y_c^{(i)} - \Omega h_{\dot{p}}^{(i)}y_s^{(i)})dt,$$

$$\hat{\sigma}_p = \frac{1}{N}\sum_{i=1}^{N}\int_0^{T_i}(\bar{h}_p^{(i)}z_c^{(i)} - \Omega\bar{h}_{\dot{p}}^{(i)}z_s^{(i)})dt$$

for $p = y$ or $z$, and where $h_q^{(i)}$ is the partial derivative of $h_1(t)$ with respect to variable $q$ and $X(\tau)y_{s,c}^{(i)}$ and $X(\tau)z_{s,c}^{(i)}$ are the $\sin(\Omega t + \phi(\tau))$ and $\cos(\Omega t + \phi(\tau))$ components of $y_1(t)$ and $z_1(t)$, respectively.

The right-hand sides of the Eqs. (23) and (24) describe the slow adaptation to the frequency and damping of the bridge due to the presence of the pedestrians. Each of these right-hand sides has three components. These represent respectively: (I) adaptation due to direct dependence of the foot force $H$ on the bridge motion, neglecting any change in timing of footsteps (the terms $\hat{h}_x$ and $\hat{h}_u$); (II) the component at the bridge frequency that is present in the adjustment to the pedestrian lateral foot placement (the terms $\hat{\kappa}_y$ and $\hat{\sigma}_y$); and (III) the component at the bridge frequency that is present in the adaptation to the pedestrian's forward motion (the terms $\hat{\kappa}_z$ and $\hat{\sigma}_z$).

Let us examine the damping Eq. (24). Note that the term of the right-hand side is the $\mathcal{O}(\varepsilon)$-component of the total negative damping of the bridge. That is, in the notation of (8)

$$\bar{\sigma}_1 = -\hat{h}_u, \quad \bar{\sigma}_2 = \frac{\hat{\sigma}_y}{\Omega}, \quad \bar{\sigma}_3 = \frac{\hat{\sigma}_z}{\Omega}.$$

Note that $\bar{\sigma}_1$ is identical to the condition derived in refs. [18,31] and expressed analytically in ref. [47] for the negative damping contribution for Model 1. The terms

$\bar{\sigma}_2$ and $\bar{\sigma}_3$ are other terms that should be considered at the same order for a general foot-force model.

**Numerical implementation**. Parameters that characterise pedestrians walking frequencies were chosen in a biomechanically realistic range. Bridge parameters were chosen close to those of the London Millennium Bridge. Table 3 contains the specific values and their sources.

Numerical simulations were performed using bespoke software written by us, mostly in Python, with some use of MATLAB and Java. Discretisation was performed using a Runge–Kutta method. Further details of the integrals underlying the computation of $\bar{\sigma}_{1,2,3}$ are contained in the Supplementary Information.

## Data availability
The data that support the findings of this study (essential code for reproducing all numerical simulations) are available online at https://doi.org/10.5281/zenodo.5042706[62].

## Code availability
Code for generating the figures and animation is also available online at https://doi.org/10.5281/zenodo.5042706[62].

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

## Acknowledgements

This work was supported by the U.S. National Science Foundation under grant No. DMS-1909924 (to I.B., K.D., and R.J.), the Ministry of Science and Higher Education of the Russian Federation under grant No. 0729-2020-0036 (to I.B.) and by the Polish National Agency for Academic Exchange (NAWA) under grant No. PPN/PPO/2019/1/00036 (to M.B.).

## Author contributions

I.B. conceived of and led the study; A.R.C. derived the generalised stability formula and wrote the first draft of the paper; R.J. performed preliminary computations; K.D. extended these to produce all numerical results in this paper; M.B. led the historical review of bridge instabilities; J.H.G.M. derived models and provided expertise on bridge engineering and dynamics; A.M. reviewed all modelling, provided expertise on bridge instability mechanisms and derived explanations for observed pedestrian behaviour.

## Competing interests

The authors declare no competing interests.

## Additional information

**Peer review information** *Nature Communications* thanks Anthony Blakeborough and the other anonymous reviewer(s) for their contribution to the peer review this work. Peer reviewer reports are available.

