## [Peer Review File · Nature Communications]

Reviewers' Comments:

Reviewer #1:

Remarks to the Author:

General comments:

This paper makes a persuasive argument that the lateral wobbling of London's Millennium Bridge on its opening day was initiated by a negative damping effect generated by individual pedestrians (via 'amplitude-modulated gait width'), and was not due to synchronization of the pedestrians' footfalls. As the authors state in the abstract, "synchronisation of pedestrians' foot placement is a consequence of, not a cause of, the instability." To support this claim, the authors present an interesting compilation of nearly 30 case studies of bridges that experienced pedestrian-induced lateral oscillations. The same lesson emerges from several theoretical and numerical modelling studies over the past decade, all of which show that synchronisation is not the essential trigger of the instability.

The authors emphasize that in these models, uncorrelated motion of pedestrians does not lead to the expected cancellation of the sideways forces they exert on the bridge as they walk. Instead, an effective negative damping can occur as pedestrians pump energy into the bridge, either as a result of their foot placement on the moving deck (as in Figure 1) or by changing the way they walk as the platform moves laterally. As the authors note, these ideas have been around in some form for quite a while. Starting with the remarks of Josephson (ref. [26], mentioned on p.3), as well as early insights from Barker [27] and the Arup team led by Dallard [11, 21], and then in a series of papers by MacDonald and co-authors, several earlier authors have emphasized that pedestrians can produce an effective negative damping.

So it was a little unclear to me what the main novelty of the current paper is. The forcefulness of the argument against the 'synchronisation myth' seems to be one important point. Another is the unified calculation (for any foot force model; three are studied in detail here) of the critical number of pedestrians and the effective negative damping per pedestrian. This calculation is mainly confined to the supplementary information, with some highlights featured in the methods section. I must admit I found this calculation difficult to follow. In any case, the asymptotic theory was tested against an ensemble of simulations of various models with various parameter values, and shown to agree reasonably well. It also appears consistent with real data on N_{crit} for the Millennium Bridge, though it's not clear to me how stringent a test this is, given the uncertainty in some of the parameters in the models. Does the model make correct predictions for other cases of lateral oscillations, in those cases where the bridge parameters and pedestrian parameters are known with some degree of confidence? Or is there still too much uncertainty in real pedestrians' foot-placement control laws to be able to say anything quantitative here?

Minor comments:

1) In Table 1, the authors state that there is "no direct evidence" that pedestrian sync occurred on London's Millennium Bridge on its opening day. This could be confusing to readers who have seen the BBC video of opening day, which showed dozens of people rocking from side to side in unison as the bridge wobbled.

2) The caption to figure 1 mentions "dashed positions" (where each pedestrian places his or her stance foot) but no dashed positions were apparent in the figure.

Reviewer #2:

Remarks to the Author:

Summary. Using numerical simulations and mathematical analyses using perturbation methods, the authors suggest that it may be possible to excite lateral bridge vibrations by a crowd of

pedestrians, without the crowd synchronizing their steps.

Major remarks:

Unfortunately, despite the authors' claims implicitly that they have shown for the first time that pedestrian synchrony is not necessary for lateral bridge vibration using a model, it seems to me that this result was established in a previous article (cited briefly by the authors), unless I misunderstand the issues. Details below.

The authors state briefly in Line 38: "Although, since then, a number of publications have cast doubt on this explanation [13–17]." (meaning, the theory that synchrony is necessary).

While reference 13 and 15 provide observational or experimental accounts of asynchrony and reference 16 is a literature review, reference 17, titled "Walking crowds on a shaky surface: stable walkers discover Millennium Bridge oscillations with and without pedestrian synchrony" described a 3D mathematical model of a biped, showing that there is a range of parameter values for which synchronization is not necessary for shaking the bridge at steady state. Specifically, it is stated in reference-17's abstract: "We simulate multiple such stable bipeds walking simultaneously on a bridge, showing that they naturally synchronize under certain conditions, but that synchronization is not required to shake the bridge."

I recommend the authors to compare their results with figure 2 of reference-17, which shows how there is a range of pedestrian numbers that results in a quasi-periodic motion (ie without phase locking), where an order parameter (measure of synchrony) is plotted to show that synchrony is not necessary for substantial bridge excitation, analogous to figure 3 and 5 of the manuscript under review. I would also be interested in understanding the differences of the simulation results with Belykh et al (2017) where they showed a regime without phase locking (see more details below).

To be clear, the authors' work presents quite interesting new mathematical analyses of the biped-bridge interaction mathematical models, using the well known method of multiple scales. However, the grander claims of a "new paradigm" (as in title) and "finally dispelling the myth" (line 20), etc. seems over-stated and does not seem justified. The title, the abstract, the introduction and the conclusions seem a bit over-stated along these lines.

More specific comments:

- Table 1 is a fantastic summary.

- It was not immediately clear to me what the authors' mathematical analyses says or assumes about the quasiperiodic regime or when there is bridge oscillation without synchrony, and how these analytic expressions compare with experiment in this non-synchronous regime. Could the authors plot the 'method of multiple-scales' version of the solutions to compare with the numerical solutions in Figures 3-5? Given that the mathematical analyses (rather than the phenomenological simulations) seem to be the main advance, it may be good to expand on how the math explains for the simulation results.

- It may be beneficial to the reader to learn more explicitly about how the results here are different from that of the manuscript in reference-19: "Foot force models of crowd dynamics on a wobbly bridge" by Belykh et al (2017), which did not seem to have seem asynchrony or quasiperiodicity, although do see some lack of phase locking in some regimes. The authors say that Model 3 is from that paper: if so, what explains the differences in the simulation results, if any? Please carefully delineate your advance versus what exists in such prior work.

Line 20. " After a careful review of the observational, experimental, and modeling evidence we finally dispel the synchronization myth and show that increased synchronization of pedestrians' foot placement is a consequence of, not a cause of the instability. "

That the initial growth in bridge amplitude need not be accompanied by synchrony seems established in Figure 2 of reference-17. Please carefully delineate your advance versus what exists

in such prior work.

- In their Discussion section, the authors may want to draw attention to the biomechanical literature on walking, that there is a vast tradition of understanding human balance through various perturbations including occasionally responses to walking surface motions or visual scene perturbations, both of which may be relevant here. For instance, see:

McAndrew, Patricia M., Jonathan B. Dingwell, and Jason M. Wilken. "Walking variability during continuous pseudo-random oscillations of the support surface and visual field." *Journal of biomechanics* 43.8 (2010): 1470-1475.

Peters, Brian T., Rachel A. Brady, and Jacob J. Bloomberg. "Walking on an oscillating treadmill: strategies of stride-time adaptation." *Ecological Psychology* 24.4 (2012): 265-278.

- The authors may want to mention that they have tried to model the human in the simplest possible manner, mainly focusing on sideways stepping dynamics, largely eschewing modeling actual forward walking and certainly have not attempted to model the full 3D motion of the human in any detail. This is fine in the spirit of simplicity, but should be made explicit.

- The above remark is especially relevant in the context of Figures 1 and 2 in the manuscript and the visualizations in the animation (Supplementary Animation), all of which show a complex 3D human animation. Such 3D visualizations are VERY far removed from a point-mass stepping sideways. Thus these 3D visualizations are misleading and seem uncharacteristic of typical applied math convention. The animation especially has no relation to the simple mathematical models used, as far as I can tell as it shows rather complex movement of the limbs and torso, none of which are modeled in the manuscript. I recommend replacing these 3D figures with stick figures more representative of the model being used -- analogous to those in the authors' previous manuscripts on this topic.

- The authors present the video shot in Nepal as evidence for the lack of synchrony and in support of the model. However, there are multiple issues with using this video as evidence, even anecdotal evidence.

First, one of the videos was shot by a person holding a camera while walking on the bridge, so its hard to disambiguate the cameraperson's movement from the bridge movement; in the other two videos as well, there is considerable camera shaking, which makes it hard to observe bridge movement. To the extent that the bridge movement can be observed, a lot of it seems vertical in response to the pedestrians walking. There is a lot of wind (as can be seen from the fluttering flags), which makes it unclear if any of the small sideways bridge oscillations are due to wind or due to the pedestrians. Finally, in all these bridge vibration phenomena, there is a simpler source of non-steadiness not present in any of the models -- especially made clear by the Nepal video. The bridge motion is different at different points along the bridge. Its vibration modes may be complex, not just sideways. So the human movement, even if was synced to the bridge movement, would need to be non-steady, changing depending on where the human is along the bridge. The human's effect on the bridge may also be different at different points on the bridge. I recommend discussing these points in the Discussion.

- I think some further mathematical detail could be provided for the three models, perhaps in the supplementary appendix. For instance, it is not explained how equation 14 is obtained.

Reviewer #3:

Remarks to the Author:

Key Results

The authors draw together a large body of research into the interaction of pedestrians and structures moving lateral to the direction of walking. Combining this with the results of simulations and analytical work they conclude the following points.

i) Pedestrian induced lateral oscillation of footbridges is not unique to the Millennium Footbridge. Cases have been reported across the world and it can have serious consequences for the

serviceability of the footbridges affected. Signs of synchronous behaviour in these cases is scarce.

- ii) The primary mechanism for the onset of pedestrian induced lateral vibrations of the Millennium Footbridge around the time of its opening was caused by a reduction in the overall system damping because each pedestrian added negative damping to the system and not the popularly held belief that the vibrations were caused by the pedestrians all walking in step. Any synchronisation that does occur is a consequence of the increased motion and not its cause.
- iii) The major component of the change in damping arises from the simplest basic walking mechanism, namely that between footfalls the lateral motion of the pedestrian can be modelled as the toppling of an inverted pendulum. The toppling is arrested after a regular interval by switching to the other foot which is placed so that the lateral motion will reverse so the pedestrian does not fall over.
- iv) Other factors affecting the foot placement, the variation in timing due to changes in step length and forward velocity, are of secondary importance. These conclusions are supported by a new perturbation analysis of the effects of pacing variations.

Validity

The main findings are based on analyses of the interaction of increasing numbers of pedestrians placed on a simple one degree of freedom model of the bridge. Three pedestrian models are used. Two are based on the well accepted placement strategy due to Hof, the first assumes that the pedestrian walks at the same pace and the second is a new modified version that adjusts the step timing to take account of changes in gait due to bridge motion. The third is a model proposed by the authors and adopts a different strategy which has a greater dependence on the bridge motion and a greater tendency to synchronise. All these models are simple but show realistic behaviour. The simulations were primed with data to replicate the Millennium Footbridge around its opening and the results reproduce the observed behaviours of the bridge including the onset of unstable oscillations. It seems that the basic argument behind the paper is sound.

The models are sound in themselves but they do not cover every possibility. They are fine for identifying the essentials of interaction between the pedestrian and the bridge. However, interactions between pedestrians is only through the behaviour of the bridge, and other possible interactions between the pedestrians, for example adaption necessary to accommodate each other's movement or changes in visual field are excluded. As a result, this paper can only show that the proposed mechanism is the probable mechanism. By excluding other routes to coherence it cannot quite kill off the opposing view.

Significance

These results are likely to be significant in the field of pedestrian bridge design in that the unstable bridge oscillations are primarily due to negative damping introduced by pedestrians (for certain combinations of pacing and structural frequency). Calculations of the scale and sign of the damping effect introduced by the pedestrians, and the consequent number of pedestrians to cause unstable oscillations, show a wide variation. The average values have a similar shape to the theoretical predictions, but the scale of the variation is significant.

The wide variation in effect is a common problem in engineering, and usually design is against a realistic worst case. This would be the usual practice, but in the text the authors indicate that the outcomes depend too much on the details of the model and they cannot see it being used in formal design criteria, which is a pity.

Data and Methodology

The search for similar cases provides interesting evidence that the problem is not confined to the Millennium Footbridge. The engineering industry is pretty good at reporting odd cases, particularly where there is public concern in the behaviour of structure, and especially since the case of the Millennium Footbridge highlighted the phenomenon, so it is likely cases of large bridge oscillations get reported. It is not so certain that the synchrony would be noted, or if it were that it was actually present.

The appropriate literature is well reviewed and discussed. The choice of models is reasonable presenting a simple and a more complicated 'standard' Hof model, and a third with a significantly different control algorithm. The analysis is essentially a Monte Carlo approach, which is a well-established method.

Analytical approach

The simulation models are reasonable and the algorithms used to solve the equations are bread and

butter stuff so can be considered very reliable.

The Monte Carlo method is statistical but apart from the averaging of results in the ensembles there is little statistical analysis. The current blizzard of dots in figure 4 showing individual results only show the range of results. The addition of (say) the 95%iles would indicate useful information about the distribution of results, particularly since engineers are more likely to work with more extreme values.

I am less able to judge the validity of the perturbation analysis. I can follow the maths, which seems logical and reasonable, but it is a bit out of my field so I cannot really provide any valuable comment.

Suggested improvements

It is common in engineering to face problems where the input and the response of structures vary widely and these are dealt with in design by considering the likely worst case. Can the approach not be taken here? There is much in the paper about why you cannot apply this method to real cases. Looking at the 95%ile (say, rarer events are likely to require very many more sample runs) might be a way in.

The paper concludes by suggesting other phenomena that may be governed by similar mechanisms. It would help if the characteristic features of the method presented in the paper were identified and the how they are reproduced in the other areas given.

Clarity and context

The paper is well written in lucid English, but I think it could do with a bit more direction and signposting of the argument. At the moment it reads as a bit of a compilation with no clearly expressed overarching theme or target. The paper claims to be about the Millennium Footbridge problem and killing off the synchronous requirement as instigator, which by excluding alternative routes for pedestrians to synchronise their steps it does not quite do. It presents a case based on a series of models that provides very strong support for the view that it is the effect the pedestrians have on the damping of the system that drives the instability, but that would require much less detailed discussion on the background to the problem. It is a pity that the paper cannot begin to come to any specific advice to bridge designers.

References

The paper does reference previous literature appropriately.

Expertise

I have a background in structural dynamics and testing, and experience of human/structure interaction.

Response to referees of: *A new paradigm for emergent instability:
the real story of the London Millennium Bridge*

Igor Belykh *et al.*

July 12, 2021

We are very grateful to the Reviewers for the constructive comments and time. These courteous and thoughtful comments have greatly helped us to improve our work. We have made numerous changes to the manuscript as a result, which are delineated using a blue font in the revised version. Please find below detailed responses to each of the referees including responses to all of their comments.

Reply to Reviewer 1

General comments:

This paper makes a persuasive argument that the lateral wobbling of London’s Millennium Bridge on its opening day was initiated by a negative damping effect generated by individual pedestrians (via ‘amplitude-modulated gait width’), and was not due to synchronization of the pedestrians’ footfalls.

As the authors state in the abstract, “synchronisation of pedestrians’ foot placement is a consequence of, not a cause of, the instability.” To support this claim, the authors present an interesting compilation of nearly 30 case studies of bridges that experienced pedestrian-induced lateral oscillations. The same lesson emerges from several theoretical and numerical modelling studies over the past decade, all of which show that synchronisation is not the essential trigger of the instability.

The authors emphasize that in these models, uncorrelated motion of pedestrians does not lead to the expected cancellation of the sideways forces they exert on the bridge as they walk. Instead, an effective negative damping can occur as pedestrians pump energy into the bridge, either as a result of their foot placement on the moving deck (as in Figure 1) or by changing the way they walk as the platform moves laterally. As the authors note, these ideas have been around in some form for quite a while. Starting with the remarks of Josephson (ref. [26], mentioned on p.3), as well as early insights from Barker [27] and the Arup team led by Dallard [11, 21], and then in a series of papers by MacDonald and co-authors, several earlier authors have emphasized that pedestrians can produce an effective negative damping.

So it was a little unclear to me what the main novelty of the current paper is. The forcefulness of the argument against the ‘synchronisation myth’ seems to be one important point. Another is the unified calculation (for any foot force model; three are studied in detail here) of the critical number of pedestrians and the effective negative damping per pedestrian.

The authors are grateful for the Reviewer’s comments, which we believe summarise well the significant contribution of this paper. The Reviewer questions though what is the key novel contribution of the paper. We accept that we may not have made this point strongly enough in the Abstract and opening paragraphs. We have made adjustments to the text there and especially in the Discussion to try to make our key novel contribution plain.

In short, the referee rightly points out that previous work has shown synchronisation is not essential to trigger instability of the London Millennium Bridge and similar structures, and that pedestrians can effectively act as negative dampers. But despite these works, at best the question is referred to as a *debate* in the literature between negative damping and synchronisation hypotheses. At worst, and in numerous presentations in print, film, radio etc., there is a wide belief that the instability that day was **caused** by a textbook example of synchronisation of coupled pedestrians - in effect “naive engineers” were put in their place by “clever physicists and mathematicians”. As we are sure this Reviewer knows, not only is this belief misguided, but (see answer below) we would argue that an overconfident belief in a synchronisation explanation is also potentially dangerous.

The main novelty of the paper then is to show, in a generic way, that the negative damping is essential for the instability and that when synchronisation occurs it is a consequence, rather than a cause, of the instability. This is achieved through asymptotic analysis applicable to any foot force model, and is demonstrated using three specific models, one of which cannot synchronise, one that includes adaptation that permits synchronisation, and the other of which is highly prone to synchronisation. The findings are supported by data and observations from a large number of real bridges that have experienced large amplitude lateral pedestrian-induced vibrations, in which direct evidence of synchronisation is at best scant.

That the problem is subtle is something that we have tried to emphasise. Synchronisation (or more precisely, increased coherence of pedestrian foot placement) can occur, especially if pedestrians happen to be walking close to the bridge’s natural frequency. But even in those cases, we show that negative damping can still be regarded as being the trigger. Also, the quantitative prediction of instability thresholds relies on the details of the pedestrian gait; clearly different individuals adopt different gaits. Nevertheless, we claim we have presented more than sufficient evidence to finally dispel the myth that synchronisation is the fundamental cause of pedestrian-induced bridge instability. It is not. In fact, we argue that the paradigm of instability due to negative damping on average, **without temporal coherence** is a simpler, yet often overlooked mechanism for emergent instability, beyond the particular problem of pedestrian bridges. Hence the title of the paper.

This calculation is mainly confined to the supplementary information, with some highlights featured in the methods section. I must admit I found this calculation difficult to follow.

Thank you for this comment. The details of the calculation are technical and hence are relegated to the supplementary information. Nevertheless, we believe that the concept of the calculation is in fact straightforward and uses a tried-and-tested methodology. We apologise that we did not do a better job of giving an overview of the calculation in the Methods section. Therefore, in response to this comment we have added a paragraph of text at the start of the “Asymptotic derivation of damping criterion” subsection of the Methods section.

In any case, the asymptotic theory was tested against an ensemble of simulations of various models with various parameter values, and shown to agree reasonably well. It also appears consistent with real data on N_{crit} for the Millennium Bridge, though it’s not clear to me how stringent a test this is, given the uncertainty in some of the parameters in the models. Does the model make correct predictions for other cases of lateral oscillations, in those cases where the bridge parameters and pedestrian parameters are known with some degree of confidence? Or is there still too much uncertainty in real pedestrians’ foot-placement control laws to be able to say anything quantitative here?

Thank you. These comments touch upon an important point: even when much is known about the physical properties of a bridge, knowledge of the crowd behaviour is necessarily subject to large uncertainties, both aleatoric and epistemic. For example, not only will there be a distribution of foot-placement control laws amongst the individuals in any crowd, but that distribution is not known. Despite this inevitable uncertainty, it is still possible to make quantitative statements. A specific point is that bridges with low natural frequencies (close to say 0.4Hz, which is much lower than the dominant lateral excitation frequency, circa 1 Hz) would not be expected to be excited by a crowd according to the main synchronisation hypothesis, since it is arguably unlikely that an individual would slow the cadence of their footfalls by a factor of 2.5 to synchronise. If that were accepted, bridge designers could thus argue that low frequency bridges need take no precautions against the possibility of lateral excitation phenomena, whereas the models analysed here show that this is far from the case. Preventative measures such as tuned mass dampers are expensive, and there are incentives for arguing that they are not necessary; our work shows that **this would be a dangerous path to take**. This paper’s demonstration of the alternative paradigm shows that the frequency range of concern is much wider than implied by some earlier theories, and the inherent uncertainties make this frequency range wider yet. Note how our scatter plots of Figure 4 provide quantitative illustrations of this.

Minor comments:

1. *In Table 1, the authors state that there is “no direct evidence” that pedestrian sync occurred on London’s Millennium Bridge on its opening day. This could be confusing to readers who have seen the BBC video of opening day, which showed dozens of people rocking from side to side in unison as the bridge wobbled.*

This is an important point, which we have now clarified in the revision, in the observational review subsection of the Results.

We encourage the Reviewer (and now following this change in the text) the reader to look again at the videos which are widely available on the Internet. A distinction needs to be made between synchronisation of head and upper body movements (readily seen in videos) and synchronisation of footfalls on the deck. We are not aware of any video footage that establishes that footfall synchrony occurred. Moreover, a walker providing an effective negative damping force to the bridge, necessarily at the bridge frequency, will exhibit a component of upper body motion at that frequency.

2. *The caption to figure 1 mentions “dashed positions” (where each pedestrian places his or her stance foot) but no dashed positions were apparent in the figure.*

Thank you for spotting this. We have changed this description to “light blue and light red positions.”

Reply to Reviewer 2:

Major remarks:

Unfortunately, despite the authors' claims implicitly that they have shown for the first time that pedestrian synchrony is not necessary for lateral bridge vibration using a model, it seems to me that this result was established in a previous article (cited briefly by the authors), unless I misunderstand the issues. Details below. The authors state briefly in Line 38: "Although, since then, a number of publications have cast doubt on this explanation [13–17]." (meaning, the theory that synchrony is necessary). While reference 13 and 15 provide observational or experimental accounts of asynchrony and reference 16 is a literature review, reference 17, titled "Walking crowds on a shaky surface: stable walkers discover Millennium Bridge oscillations with and without pedestrian synchrony" described a 3D mathematical model of a biped, showing that there is a range of parameter values for which synchronization is not necessary for shaking the bridge at steady state. Specifically, it is stated in reference-17's abstract: "We simulate multiple such stable bipeds walking simultaneously on a bridge, showing that they naturally synchronize under certain conditions, but that synchronization is not required to shake the bridge."

The authors are grateful for the Reviewer's comments. The paper makes clear that earlier papers, including a number by the authors themselves, have presented evidence - both mathematical and experimental (in laboratories and at full-scale) - for lateral bridge excitation by asynchronous walkers. All of these works show examples where synchronisation is not a necessary ingredient of instability. This includes reference 17, which we have already cited. We agree that that was a key paper that makes a worthwhile contribution, but we feel that the Referee has not understood the main novelty and greater generality of what has been shown here (see also comments in answer to the Reviewer's more specific points below). In particular, reference 17 only presents simulation results for a particular pedestrian model. It also only deals with the case of mild detuning of the pedestrians from each other's walking frequencies. It does not provide any theoretical calculations to support its conclusion for other walking models (in fact it uses an energy optimised model and in reality pedestrians adopt a wide variety of gaits). Nor does that paper provide any systematic review of observational and experimental results.

In contrast, the main novel claim of the present paper is to show, in a generic way, that the negative damping (on average) effect of any pedestrian's walking, at **any stride frequency** is essential for the pedestrian-induced instability. Moreover, when synchronisation occurs it is a consequence, rather than a cause, of the instability. This is achieved through asymptotic analysis applicable to any foot-force model, and is demonstrated using three specific models, one of which cannot synchronise, one that includes adaptation that permits synchronisation, and the other of which is highly prone to synchronisation. The findings are supported by data and observations from a large number of real bridges that have experienced large amplitude lateral pedestrian-induced vibrations, in which direct evidence of synchronisation is at best scant.

Another principal new finding of our paper is that, for the first time, we demonstrate that pedestrians that are capable of phase-locking (Model 2) or have a strong propensity for synchronisation (Model 3) induce bridge instability **prior** to the onset of crowd synchrony. This result carries over to the worst scenario case of identical pedestrians as evidenced in the simulations given in the Supplementary Information. Our finding - that synchronisation is a consequence not a cause of bridge wobbling - has not been reported in any of the above referenced papers. It represents a new and surprising result that challenges the synchronisation hypothesis, even in the case where pedestrians are prone to strong phase-locking.

I recommend the authors to compare their results with figure 2 of reference-17, which shows how there is a range of pedestrian numbers that results in a quasi-periodic motion (ie without phase locking), where an order parameter (measure of synchrony) is plotted to show that synchrony is not necessary for substantial bridge excitation, analogous to figure 3 and 5 of the manuscript under review. I would also be interested in understanding the differences of the simulation results with Belykh et al (2017) where they showed a regime without phase locking (see more details below).

We are grateful for this recommendation, but, the statement of the first sentence - “. . . there is a range . . .” – appears to be a little too definitive and possibly over-stated, since reference 17 reports only on a mathematical model, not necessarily reality. Nevertheless we are happy to comment.

There is indeed some similarity of Figure 2 of reference 17 to our Figures 3 and 8 (not Figure 5), but the key features we have shown, that have not been identified before, are (i) that the instability is fully explained by the generalised negative damping analysis, for any foot force model - the amplitude of vibrations grow exponentially as soon as the damping becomes negative. And (ii) that synchronisation, when it occurs, does so **after** the initiation of the large vibrations. We have made these points somewhat clearer now in a revised Discussion section.

The comment on the differences of the simulation results from Belykh et al. (2017) is addressed below.

To be clear, the authors’ work presents quite interesting new mathematical analyses of the biped-bridge interaction mathematical models, using the well known method of multiple scales. However, the grander claims of a “new paradigm” (as in title) and “finally dispelling the myth” (line 20), etc. seems over-stated and does not seem justified. The title, the abstract, the introduction and the conclusions seem a bit over-stated along these lines.

As we noted in previous comments, it is easy to appear to over-state matters. However, there was a deliberate choice to state the findings clearly, forcefully and unambiguously, given the overwhelming prevalence of the synchronisation theory in some fields, such as the nonlinear dynamics community and, as a consequence, the allied fields of experimental and theoretical physics. Researchers whose backgrounds are more centred on human and animal locomotion or even bridge design may be more aware of the alternative explanations, and less aware of how absent those explanations are in other fields. Nevertheless, we have adjusted the text in the Abstract, Introduction and Discussion.

More specific comments:

Table 1 is a fantastic summary.

Many thanks.

It was not immediately clear to me what the authors’ mathematical analyses says or assumes about the quasiperiodic regime or when there is bridge oscillation without synchrony, and how these analytic expressions compare with experiment in this non-synchronous regime.

The main result of the analysis is that all of the large lateral bridge vibrations are due to negative damping, whether or not there is synchrony (or quasiperiodic behaviour). Negative damping is the common factor, with synchrony being a secondary effect which occurs in some cases, after the initial instability, even for Model 3 which is prone to synchronisation. This has been clarified in the Discussion.

As regards experiments, data from independent laboratory experiments on a laterally oscillating walkway and three different laterally oscillating treadmills, in all of which synchronisation was rarely observed, have been found to exhibit similar features to those from the inverted pendulum model (see references 15, 16, 32, 33 and 75), as mentioned under Model 1 in the Methods section.

Could the authors plot the 'method of multiple-scales' version of the solutions to compare with the numerical solutions in Figures 3-5? Given that the mathematical analyses (rather than the phenomenological simulations) seem to be the main advance, it may be good to expand on how the math explains for the simulation results.

The effective damping plotted in Fig. 3-5 is actually calculated through the general expression (1) derived via the “method of multiscales,” thereby validating the predictive power of our analysis. To make it clearer, we have added clarifications to the main text and the caption of Fig. 4.

It may be beneficial to the reader to learn more explicitly about how the results here are different from that of the manuscript in reference-19: "Foot force models of crowd dynamics on a wobbly bridge" by Belykh et al (2017), which did not seem to have seen asynchrony or quasiperiodicity, although do see some lack of phase locking in some regimes. The authors say that Model 3 is from that paper: if so, what explains the differences in the simulation results, if any? Please carefully delineate your advance versus what exists in such prior work.

The simulations of pedestrian-bridge interactions described by a van der Pol-type model and Model 3 performed in Belykh et al. [19] indicate that the onset of bridge oscillations is generally accompanied by the emergence of crowd synchrony. See Figs. 4 and 8 in [19]. Except for some minor phase detuning, these simulations supported the synchronisation hypothesis. These simulations were performed for fixed crowd sizes such that a fixed number of walkers were placed on the bridge and the walker-bridge system was integrated for a long time thereby allowing the system to settle down to an established regime. Then, the crowd size was increased, and the simulations were repeated again. This setup seems to be identical to that of Ref. [17]. On the contrary, the simulations in the present paper are aimed to more closely replicate the set of controlled experiments on the London Millennium Bridge prior to reopening [11,21]. In those experiments pedestrians were added gradually (in groups of 10) so that the crowd size increased as a function of time. This was also the main setup for simulations in Strogatz et al, 2005 [12] which suggested that bridge instability was caused by crowd synchrony. The key difference between the results of [19] and our paper is that despite the strong propensity of Model 3 for synchronization as shown in [19], our results here show onset of bridge instability *prior* to the onset of crowd synchrony in the targeted experiment with gradually increasing crowd size.

We have added a clarifying paragraph to “Simulation results.”

Line 20. " After a careful review of the observational, experimental, and modeling evidence we finally dispel the synchronization myth and show that increased synchronization of pedestrians' foot placement is a consequence of, not a cause of the instability. " That the initial growth in bridge amplitude need not be accompanied by synchrony seems established in Figure 2 of reference-17. Please carefully delineate your advance versus what exists in such prior work.

We agree that it is well known that the growth of bridge amplitude need not necessarily be accompanied by synchrony, as discussed in the introduction to our paper. This was originally modelled in Barker, 2002 [27] and developed in Macdonald, 2009 [30], then again shown in Joshi and Srinivasan, 2018 [17], each for a specific model. However, we have now shown, in a generic

way, that the negative damping is essential for the instability, whether or not there is synchrony, and that when synchrony occurs it is a consequence, rather than the cause of the instability.

Moreover, as stated in Ref. [17], Fig. 2 of that paper is for the restricted case of walkers whose footfall frequencies are close to the bridge's natural frequency. From the perspective of designing safe bridges, it is well-known that large excitations can occur in such conditions, thus drawing distinctions between the gait micromechanics there, as Ref. [17] does, is of arguably of less interest to bridge designers than the current paper's analyses over the full range of bridge frequencies. Ref. [17] states that it uses "a spring and damper with values corresponding to the London Millennium Bridge," yet the London Millennium Bridge has three spans and numerous relevant modes. The Supplementary Information accompanying Ref. 17 states use of $M = 1.13 \times 10^5$ kg and $K = 4.73 \times 10^6$ Nm¹, suggesting a bridge frequency of $f = 1.03$ Hz. As previously stated, the results in the current paper that will be of greatest interest to bridge designers will be those with bridge frequencies well away from 1 Hz.

In summary, while many papers have shown that negative damping can be an explanation of instability, we stand by the claims that the present paper shows, for the first time, that negative damping is necessary in the general case and that even when some level of synchronisation occurs it is a consequence, rather than the cause, of bridge vibrations.

In their Discussion section, the authors may want to draw attention to the biomechanical literature on walking, that there is a vast tradition of understanding human balance through various perturbations including occasionally responses to walking surface motions or visual scene perturbations, both of which may be relevant here. For instance, see:

McAndrew, Patricia M., Jonathan B. Dingwell, and Jason M. Wilken. "Walking variability during continuous pseudo-random oscillations of the support surface and visual field." Journal of biomechanics 43.8 (2010): 1470-1475.

Peters, Brian T., Rachel A. Brady, and Jacob J. Bloomberg. "Walking on an oscillating treadmill: strategies of stride-time adaptation." Ecological Psychology 24.4 (2012): 265-278.

The authors are grateful for these suggestions. These references have now been included in the methods section where we introduce the models.

The authors may want to mention that they have tried to model the human in the simplest possible manner, mainly focusing on sideways stepping dynamics, largely eschewing modeling actual forward walking and certainly have not attempted to model the full 3D motion of the human in any detail. This is fine in the spirit of simplicity, but should be made explicit.

The authors are grateful for this suggestion, we had already made this point "parsimonious assumption" in the main body, but we have now added an extra paragraph to made this clear in the Methods section.

The above remark is especially relevant in the context of Figures 1 and 2 in the manuscript and the visualizations in the animation (Supplementary Animation), all of which show a complex 3D human animation. Such 3D visualizations are VERY far removed from a point-mass stepping sideways. Thus these 3D visualizations are misleading and seem uncharacteristic of typical applied math convention. The animation especially has no relation to the simple mathematical models used, as far as I can tell as it shows rather complex movement of the limbs and torso, none of which are modeled in the manuscript. I recommend replacing these 3D figures with stick figures more representative of the model being used – analogous to those in the authors' previous manuscripts on this topic.

We have adapted the animation by including a two-legged inverted pendulum and removing the apparent upper motion of its 3D humanoid avatar. We have also made it clear in the description exactly how the simulations were made.

The authors present the video shot in Nepal as evidence for the lack of synchrony and in support of the model. However, there are multiple issues with using this video as evidence, even anecdotal evidence. First, one of the videos was shot by a person holding a camera while walking on the bridge, so its hard to disambiguate the cameraperson's movement from the bridge movement; in the other two videos as well, there is considerable camera shaking, which makes it hard to observe bridge movement. To the extent that the bridge movement can be observed, a lot of it seems vertical in response to the pedestrians walking. There is a lot of wind (as can be seen from the fluttering flags), which makes it unclear if any of the small sideways bridge oscillations are due to wind or due to the pedestrians. Finally, in all these bridge vibration phenomena, there is a simpler source of non-steadiness not present in any of the models – especially made clear by the Nepal video. The bridge motion is different at different points along the bridge. Its vibration modes may be complex, not just sideways. So the human movement, even if was synced to the bridge movement, would need to be non-steady, changing depending on where the human is along the bridge. The human's effect on the bridge may also be different at different points on the bridge. I recommend discussing these points in the Discussion.

We agree with the Reviewer that this video evidence for the lack of phase locking is anecdotal and there might also be other contributing factors, including (i) the mode shapes being non-uniform over the length of the bridge, implying that the bridge and trekker motion could be transient, and (ii) additional lateral bridge motion due to the wind, in which case phase locking of the bridge and trekkers would not necessarily be expected. These videos do not fully demonstrate the separation of all these confounding factors. However, thousands of Everest-bound trekkers and climbers, crossing the Hillary Bridge, named after Sir Edmund Hillary and featured in Fig. 7 b,c, would attest from their experience that (i) significant sideways bridge oscillations are rather induced by the pedestrians and not by the wind and (ii) the evidence for pedestrian synchronisation on this and other Dudh Kosi River bridges is at best scant. We have added this discussion to Supplementary Information.

I think some further mathematical detail could be provided for the three models, perhaps in the supplementary appendix. For instance, it is not explained how equation 14 is obtained.

Thank you for the suggestion. We have added some explanatory text at the introduction to that section. We have also included an intermediate step before Eq. (14) to make it clearer how the formula was obtained and we have corrected a typo in the equation for the step timing t_s , for Model 2. We have also augmented Eq. (13) for Model 1 to indicate how the alternative assumptions of step width adjustments based on relative and absolute velocities are handled. Otherwise, Model 2, which is new, is fully explained, while Model 1 is directly from references [30] and [31] and Model 3 from references [19] and [62], as cited, so we do not believe that further mathematical detail of those models is required.

Reply to Reviewer 3

We thank the Reviewer for their positive comments and overall assessment of the paper. We shall only deal with points of criticism or suggested improvements

Validity

It seems that the basic argument behind the paper is sound. The models are sound in themselves but they do not cover every possibility. They are fine for identifying the essentials of interaction between the pedestrian and the bridge. However, interactions between pedestrians is only through the behaviour of the bridge, and other possible interactions between the pedestrians, for example adaption necessary to accommodate each other's movement or changes in visual field are excluded. As a result, this paper can only show that the proposed mechanism is the probable mechanism. By excluding other routes to coherence it cannot quite kill off the opposing view.

We agree with the Reviewer that no theory can ever kill off opposing views. We also agree that, at least in theory, there are a number of mechanisms by which pedestrians could be prompted to generate synchronised loading onto the bridge. Nevertheless, recent direct evidence suggests that pedestrian-structure rather than pedestrian-pedestrian interaction is dominant in this case [2]. We have now included a paragraph at the start of the Methods section to make this point clear and to justify our choice of modelling approach. In addition to an explicit reference to [2], we also point out how visual and auditory stimuli do not lead to significant levels of spontaneous synchronisation within a group of pedestrians walking on stationary ground [5]. In contrast, from the perspective of functional human gait, unlike synchronisation, medio-lateral stability can be considered as one of the critical components of human gait [1]. Therefore, the primary objective of pedestrians walking on vibrating ground is to remain balanced, by adjustment of step width, for which there is direct evidence [3, 4].

We have adjusted the discussion at the start of the Methods section to include these points and new references.

These results are likely to be significant in the field of pedestrian bridge design in that the unstable bridge oscillations are primarily due to negative damping introduced by pedestrians (for certain combinations of pacing and structural frequency). Calculations of the scale and sign of the damping effect introduced by the pedestrians, and the consequent number of pedestrians to cause unstable oscillations, show a wide variation. The average values have a similar shape to the theoretical predictions, but the scale of the variation is significant. The wide variation in effect is a common problem in engineering, and usually design is against a realistic worst case. This would be the usual practice, but in the text the authors indicate that the outcomes depend too much on the details of the model and they cannot see it being used in formal design criteria, which is a pity.

The variation in the results is indeed significant. Assumptions are made in the different models and there are uncertainties in some of the parameter values. But even for a given model for given parameter values, there is a lot of scatter in the simulated results, e.g. relative to the closed form averaged solution for Model 1 [63] (Figure 4(left)). This scatter is due to the random nature of the pedestrian loading - over short times random left and right foot forces can feed energy into or extract energy from the bridge. Further work, especially experimental, is required to establish a quantitatively reliable model and to characterise the variations. However, qualitatively we have demonstrated that the negative damping effect is fundamental to the instability and that

the range of bridge frequencies over which the instability can occur is greater than previously appreciated. This justifies the use of formal design criteria based on negative damping, rather than other approaches, though the specific values and frequency-dependency are yet to be refined.

Data and Methodology

The search for similar cases provides interesting evidence that the problem is not confined to the Millennium Footbridge. The engineering industry is pretty good at reporting odd cases, particularly where there is public concern in the behaviour of structure, and especially since the case of the Millennium Footbridge highlighted the phenomenon, so it is likely cases of large bridge oscillations get reported. It is not so certain that the synchrony would be noted, or if it were that it was actually present.

We agree that the presence or absence of pedestrian synchrony on bridges is less likely to be reported than large vibrations, but the data in Table 1 shows that evidence of synchronisation, which is often assumed to be the cause of large lateral vibrations, is in fact very limited.

The simulation models are reasonable and the algorithms use to solve the equations are bread and butter stuff so can be considered very reliable. The Monto Carlo method is statistical but apart from the averaging of results in the ensembles there is little statistical analysis. The current blizzard of dots in figure 4 showing individual results only show the range of results. The addition of (say) the 95iles would indicate useful information about the distribution of results, particularly since engineers are more likely to work with more extreme values.

This is a good point. We have performed additional statistical analysis of the data in Fig. 4, have replotted the data, and added 5th percentile curves. These curves indicate that negative damping can be observed at any (not necessarily resonant) frequency in the considered range of frequency ratios.

Suggested improvements

It is common in engineering to face problems where the input and the response of structures vary widely and these are dealt with in design by considering the likely worst case. Can the approach not be taken here? There is much in the paper about why you cannot apply this method to real cases. Looking at the 95%ile (say, rarer events are likely to require very many more sample runs) might be a way in.

As stated above, qualitatively the paper explains the fundamental mechanism of the instability and demonstrates that a negative damping approach is appropriate for application to real cases. Furthermore the range of bridge frequencies that are vulnerable to lateral dynamic instability is greater than previously appreciated. However, further work, especially experimental, is required to define reliable quantitative values for real application.

The paper concludes by suggesting other phenomena that may be governed by similar mechanisms. It would help if the characteristic features of the method presented in the paper were identified and the how they are reproduced in the other areas given.

We have taken the Reviewer's suggestion on board. We have included a couple more paragraphs of explanation and also an additional few references, including to a previous paper by one of us (AMcR) that considered a Phillips Machine as a model of economic cycles.

Clarity and context

The paper is well written in lucid English, but I think it could do with a bit more direction and signposting of the argument. At the moment it reads as a bit of a compilation with no clearly expressed overarching theme or target. The paper claims to be about the Millennium Footbridge problem and killing off the synchronous requirement as instigator, which by excluding alternative routes for pedestrians to synchronise their steps it does not quite do. It presents a case based on a series of models that provides very strong support for the view that it is the effect the pedestrians have on the damping of the system that drives the instability, but that would require much less detailed discussion on the background to the problem. It is a pity that the paper cannot begin to come to any specific advice to bridge designers.

We have made many minor revisions to signpost the argument. In particular, we have totally rewritten the discussion to explain why the advice to bridge designers is simply to include significant lateral damping. As stated above, further work, especially experimental, is required to define reliable quantitative relationships, such as between the required amount of damping and modal frequency, because of natural variation in human gait between walkers.

References

- [1] R.M. Alexander. *Principles of animal locomotion*. Princeton, USA, 2003.
- [2] M. Bocian, J.M.W. Brownjohn, V. Racic, D. Hester, A. Quattrone, L. Gilbert, and R. Beasley. Time-dependent spectral analysis of interactions within groups of walking pedestrians and vertical structural motion using wavelets. *Mechanical Systems and Signal Processing*, 105:502–523, 2018.
- [3] P.M. McAndrew, J.B. Dingwell, and J.M. Wilken. Walking variability during continuous pseudo-random oscillations of the support surface and visual field. *Journal of Biomechanics*, 43:1470–1475, 2010.
- [4] B.T. Peters, R.A. Brady, and J.J. Bloomberg. Walking on an oscillating treadmill: strategies of stride-time adaptation. *Ecological Psychology*, 24:265–278, 2012.
- [5] A.A. Soczawa-Stronczyk, M. Bocian, H. Wdowicka, and J. Malin. Topological assessment of gait synchronisation in overground walking groups. *Human Movement Science*, 66:541–553, 2019.

Reviewers' Comments:

Reviewer #1:

Remarks to the Author:

My comments have been addressed. But I have one last (optional) suggestion. In response to one of my comments, the authors responded more powerfully to me in their rebuttal letter than they did in the revised manuscript. Namely, when explaining why the synchronisation myth could be downright dangerous in the real world, the authors made the case very well in the following passage, which appears in their rebuttal letter. I'd suggest that they include this passage in its entirety in the manuscript, as opposed to the briefer (and less compelling) remarks that currently appear there:

``... an important point: even when much is known about the physical properties of a bridge, knowledge of the crowd behaviour is necessarily subject to large uncertainties, both aleatoric and epistemic. For example, not only will there be a distribution of foot-placement control laws amongst the individuals in any crowd, but that distribution is not known. Despite this inevitable uncertainty, it is still possible to make quantitative statements. A specific point is that bridges with low natural frequencies (close to say 0.4Hz, which is much lower than the dominant lateral excitation frequency, circa 1 Hz) would not be expected to be excited by a crowd according to the main synchronisation hypothesis, since it is arguably unlikely that an individual would slow the cadence of their footfalls by a factor of 2.5 to synchronise. If that were accepted, bridge designers could thus argue that low frequency bridges need take no precautions against the possibility of lateral excitation phenomena, whereas the models analysed here show that this is far from the case. Preventative measures such as tuned mass dampers are expensive, and there are incentives for arguing that they are not necessary; our work shows that this would be a dangerous path to take. This paper's demonstration of the alternative paradigm shows that the frequency range of concern is much wider than implied by some earlier theories, and the inherent uncertainties make this frequency range wider yet. Note how our scatter plots of Figure 4 provide quantitative illustrations of this.'

Reviewer #2:

Remarks to the Author:

It would be preferable for reviewers if the authors' response to reviewers document contained quotes from the manuscript that addressed the reviewer comments, or at least, line numbers corresponding to their changes (rather than stating that they have added some sentences in the Discussion section). This would help the reviewers better judge whether their comments have been addressed or rebutted thoughtfully.

Some of the extensive addressing of the reviewer remarks that are in the response to reviewers document do not seem to have found their way into the manuscript itself. I feel that the authors could better contrast their work in the manuscript itself, in a constructive manner, especially given their grand statements about their own work ("new paradigm", etc.). I believe this will make their scholarship/contributions stronger and allow their work to be better appreciated by the readers.

As an aside, in their 'response to reviewers', they state regarding their own manuscript (in contrast to prior work): "In contrast, the main novel claim of the present paper is to show, in a generic way, that the negative damping (on average) effect of any pedestrian's walking, at any stride frequency is essential for the pedestrian-induced instability. Moreover, when synchronisation occurs it is a consequence, rather than a cause, of the instability."

This seems to be a statement (specifically the genericity claim) that may be hard to justify by the modeling or the mathematics. The authors' critique of previous work (as a mathematical model not reality or having using a specific pedestrian model) in the response to reviewers is true of their manuscript as well -- and indeed ALL modeling studies -- that the results are contingent on the mathematical and structural assumptions embedded in the model. Even modeling the person's

effect as equivalent to the forces that they exert on the bridge based on a reduced state that consists of bridge and the person's center of mass, say, is a remarkably simplifying assumption that has nothing generic about it. Generalizing foot force models do not take away from these assumptions. The genericity may be true for a particular mathematical model and particular types of model perturbations, but may not be true for all possible models for the phenomenon or model perturbations. To be clear, I don't think there is anything wrong with such simplifying assumptions or lack of genericity, as long as grander claims are not made and the models are reasonable. And I *do* think the authors' models are reasonable.

In any case, to repeat myself, I still feel it would be better if the authors were to tone down the claims to an appropriate level, more clearly and explicitly contrasting their work in the manuscript itself, and address the limitations of their model more clearly.

The title "A new paradigm for emergent instability: the real story of the London Millennium Bridge" still seems journalistic rather than scientifically descriptive. Further, while the authors have made some solid contributions, I do not know that their contributions warrant calling them "the real story of the London Millennium Bridge" any more than the many other manuscripts they have cited.

In my previous review, I had requested toning down such rhetoric but the authors argued that they still want to retain it. I would again suggest toning down such claims and make the title more descriptive, something about negative damping, phase-locking and synchrony, their main contributions as they themselves state. I do think these contributions are interesting.

Accessibility issues with the videos.

Four videos were attached in this submission. One of the videos was not play-able on regular quicktime player version 10.5 on my macbook computer: 299741_1_video_5726949_qw41wy.mp4 (but I was able to view it after converting it using ffmpeg, so my comments below are based on this conversion).

Please check if your videos are open-able using standard software, or please suggest recommended software.

Minor comments:

The 299741_1_video_5726949_qw41wy.mp4 shows the computer generated animation of one person walking. The authors have added a point-mass version of the animation, in response to my critique that the full-body model animation was misleading to the viewer. This is much better but the point-mass model animation has the potential issue that the legs do not connect at the body center of mass, but have a finite hip widths. Is this finite hip widths true of the mathematical model? Also, analogously, the complexity of the figures 1-2 are still misleading relative to the simplicity of the model. The authors could ideally explicitly note this in at least one of the figure captions (that the real model is much simpler) and/or have the point-mass model also depicted in the figures.

Supplementary Videos:

In my previous review, I had expressed a number of concerns regarding the videos: specifically the un-controlled nature of the videos in Nepal and the fact that the human-induced aspect did not seem stark. Because of these concerns, I still believe that the videos do not add anything scientifically to the manuscript. Here are my concerns from the last version re-cappeded.

" - The authors present the video shot in Nepal as evidence for the lack of synchrony and in

support of the model. However, there are multiple issues with using this video as evidence, even anecdotal evidence. First, one of the videos was shot by a person holding a camera while walking on the bridge, so it's hard to disambiguate the cameraperson's movement from the bridge movement; in the other two videos as well, there is considerable camera shaking, which makes it hard to observe bridge movement. To the extent that the bridge movement can be observed, a lot of it seems vertical in response to the pedestrians walking. There is a lot of wind (as can be seen from the fluttering flags), which makes it unclear if any of the small sideways bridge oscillations are due to wind or due to the pedestrians. "

While the authors have added a good short paragraph in the Supplementary Information addressing the videos's limitations, they should ideally state these reservations clearly in the main manuscript. Also, I'm not sure that statements like "However, thousands of Everest-bound trekkers and climbers, 780 crossing the Hillary Bridge, named after Sir Edmund Hillary and featured in Fig. 7 b,c, would attest from their experience that ..." are admissible in a scientific article without either survey data or say a reference to a newspaper article that states something to the effect that is independent of the authors or these research). But reporting on personal experience sounds fine to me, as long as mentioned as an anecdote.

Reviewer #3:

Remarks to the Author:

Key Results

The authors draw together a large body of research into the interaction of pedestrians and structures moving lateral to the direction of walking. Combining this with the results of simulations and analytical work they conclude the following points.

- i) Although the pedestrian induced lateral oscillation of footbridges is most well-known from the case of the Millennium Footbridge at its opening, it is not a unique case. Other cases have been reported across the world leading to serious consequences for the serviceability of the footbridges affected. Analysis of the reports of these cases yields scant evidence that the pedestrians were all walking in step with each other and with the bridge.
- ii) The primary mechanism for the onset of pedestrian induced lateral vibrations of the Millennium Footbridge around the time of its opening was the reduction in the overall system damping because each pedestrian added negative damping to the system and not the popularly held belief that the vibrations were caused by the pedestrians all walking in step. A series of simulations shows that any synchronisation that does occur is a consequence of the increased motion and not its cause.
- iii) The major component of the change in damping arises from the simplest basic walking mechanism, namely that between footfalls the lateral motion of the pedestrian can be modelled as the toppling of an inverted pendulum. The toppling is arrested after a regular interval by switching to the other foot which is placed so that the lateral motion will reverse so the pedestrian does not fall over. Under certain combinations of pacing/bridge frequencies the net effect is to add or subtract energy from the bridge. It is not essential that the pacing frequency coincides with a natural frequency of the bridge, although the effect is stronger if it does.
- iv) Other factors affecting the pedestrian gait - foot placement, the variation in timing due to changes in step length and forward velocity - are of secondary importance. These conclusions are supported by a new multiple scale analysis of the effects of pacing variations.

Validity

The main findings are based on analyses of the interaction of increasing numbers of pedestrians placed on a simple one degree of freedom model of the bridge. Three pedestrian models are used. Two are based on the well accepted placement strategy due to Hof, the first assumes that the pedestrian walks at the same pace and the second is a new modified version that adjusts the step timing to take account of changes in gait due to bridge motion. The third is a model proposed by the authors and adopts a different strategy which has a greater dependence on the bridge motion and a greater tendency to synchronise. All these models are simple but show realistic behaviour. The simulations were primed with data to replicate the Millennium Footbridge around its opening and the results reproduce the observed behaviours of the bridge including the onset of unstable oscillations. It seems that the basic argument behind the paper is sound.

The models are sound in themselves but they do not cover every possibility. They are fine for identifying the essentials of interaction between the pedestrian and the bridge. However, interactions between pedestrians is only through the behaviour of the bridge, and other possible interactions between the pedestrians, for example adaptation necessary to accommodate each other's movement or changes in visual field are excluded. As a result, this paper can only show that the proposed mechanism is the probable mechanism encapsulating the essence of the physics. By excluding other routes to coherence it cannot quite kill off the opposing view, although it does seem to kill off the Kuramoto route.

Significance

These results are likely to be significant in understanding the phenomenon in the field of pedestrian bridge design. However, the calculations of the scale and sign of the damping effect introduced by the pedestrians, and the consequent number of pedestrians to cause unstable oscillations, show a wide variation. The average values have a similar shape to the theoretical predictions, but the scale of the variation is significant. The authors are pessimistic as to how this might be used in design. The walking model is very basic and reality it may be significantly more complicated. The analysis also takes no account of the effects of the mode shape of the bridge's vibration or variations in the individual mass, pace and distribution of walkers. These aspects, and no doubt others, will affect the magnitude of the effect. However, the engineering profession is well adapted to using simple models. As long as the model captures the essence of the phenomenon it can be adapted and calibrated against measured data. I see no reason why this should not be possible here. In the end though, the only way out is to add structural damping, and all a more detailed analysis would give is a more precise figure about the level that is required.

Data and Methodology

The search for similar cases provides interesting evidence that the problem is not confined to the Millennium Footbridge. The engineering industry is pretty good at reporting odd cases, particularly where there is public concern in the behaviour of structure, and especially since the case of the Millennium Footbridge highlighted the phenomenon, so it is likely cases of large bridge oscillations get reported. It is not so certain that the synchrony would be noted, or if it were that it was actually present.

The appropriate literature is well reviewed and discussed. The choice of models is reasonable presenting a simple and a more complicated 'standard' Hof model, and a third with a significantly different control algorithm. The analysis is essentially a Monte Carlo approach, which is a well-established method.

Analytical approach

The simulation models are reasonable and the algorithms used to solve the equations are bread and butter stuff so can be considered very reliable.

The Monte Carlo method is statistical and the addition of the 95 percentile adds useful information indicating the distribution of results, particularly since engineers are more likely to work with more extreme values.

I am less able to judge the validity of the multiple scale analysis. I can follow the maths, which seems logical and reasonable, but it is a bit out of my field so I cannot really provide any valuable comment here.

Clarity and context

The paper is well written in lucid English. The argument is clearly stated and follows a good logical line. It is now over 20 years since the completion of the Millennium footbridge and the events surrounding its opening. Over the intervening period there has been much discussion and, more importantly, research into the cause. This paper does outline the beginnings of a method to assess pedestrian induced lateral motion correctly. Given that it has taken so long to get to what looks like the true explanation indicates that some of the criticism heaped on the 'naïve' designers is not justified. This paper does set this aspect straight, but it is more important to note also that the engineers did fix the original problem pretty quickly.

References

The paper does reference previous literature appropriately.

Expertise

I have a background in structural dynamics and testing, and experience of human/structure interaction.

Response to referees of: *The London Millennium Bridge revisited: emergent instability without synchronisation*

September 27, 2021

This letter accompanies the submission of the revised paper. We are very grateful to the Reviewers for the constructive comments and time. These courteous and thoughtful comments have greatly helped us to improve our work. We have made numerous changes to the manuscript as a result, which are delineated using a red font in the revised version. These changes include the new title and the removal of the videos related to footbridges in Nepal. Please find below detailed responses to each of the referees including responses to all of their comments.

We trust that the revised version will now be suitable for publication.

Yours Sincerely,

Igor Belykh (on behalf of all authors).

Reviewer 1:

My comments have been addressed. But I have one last (optional) suggestion. In response to one of my comments, the authors responded more powerfully to me in their rebuttal letter than they did in the revised manuscript. Namely, when explaining why the synchronisation myth could be downright dangerous in the real world, the authors made the case very well in the following passage, which appears in their rebuttal letter. I'd suggest that they include this passage in its entirety in the manuscript, as opposed to the briefer (and less compelling) remarks that currently appear there:

"... an important point: even when much is known about the physical properties of a bridge, knowledge of the crowd behaviour is necessarily subject to large uncertainties, both aleatoric and epistemic. For example, not only will there be a distribution of foot-placement control laws amongst the individuals in any crowd, but that distribution is not known. Despite this inevitable uncertainty, it is still possible to make quantitative statements. A specific point is that bridges with low natural frequencies (close to say 0.4Hz, which is much lower than the dominant lateral excitation frequency, circa 1 Hz) would not be expected to be excited by a crowd according to the main synchronisation hypothesis, since it is arguably unlikely that an individual would slow the cadence of their footfalls by a factor of 2.5 to synchronise. If that were accepted, bridge designers could thus argue that low frequency bridges need take no precautions against the possibility of lateral excitation phenomena, whereas the models analysed here show that this is far from the case. Preventative measures such as tuned mass dampers are expensive, and there are incentives for arguing that they are not necessary; our work shows that this would be a dangerous path to take. This paper's demonstration of the alternative paradigm shows that the frequency range of concern is much wider than implied by some earlier theories, and the inherent uncertainties make this frequency range wider yet. Note how our scatter plots of Figure 4 provide quantitative illustrations of this."

Thank you for the suggestion. We have now included this paragraph in the Discussion.

Reviewer 2:

It would be preferable for reviewers if the authors' response to reviewers document contained quotes from the manuscript that addressed the reviewer comments, or at least, line numbers corresponding to their changes (rather than stating that they have added some sentences in the Discussion section). This would help the reviewers better judge whether their comments have been addressed or rebutted thoughtfully.

The new changes to the manuscript are shown in red text for them to be easily identified by the reviewers.

Some of the extensive addressing of the reviewer remarks that are in the response to reviewers document do not seem to have found their way into the manuscript itself. I feel that the authors could better contrast their work in the manuscript itself, in a constructive manner, especially given their grand statements about their own work ("new paradigm", etc.). I believe this will make their scholarship/contributions stronger and allow their work to be better appreciated by the readers.

Thank you for the suggestion. We had tried to keep the modifications to the manuscript brief, with longer justifications to try to persuade the referee of the veracity of our arguments. Given this comment though, we are happy to include more details in the manuscript. Please see the modifications to the now expanded discussion section, shown in red. We have also contrasted our work to the existing papers and added the following paragraph (lines 245-253): "Similarly, the previous work [17] studied the London Millennium Bridge instability for fixed crowd sizes. This work used an energy-optimised pedestrian model with a linear feedback controller to demonstrate that heterogeneous pedestrians incapable of synchronising even at large crowd sizes can shake the bridge without synchronisation. However, once the stride frequencies were chosen to be equal, identical pedestrians triggered the bridge instability simultaneously with the onset of crowd synchrony [17]."

On the contrary, our results indicate that pedestrians with a weak (Model 2) or strong (Model 3) propensity for synchronisation can induce the bridge vibrations prior to the onset of crowd synchronisation when added sequentially (also see the extreme case of identical pedestrians in Fig. 8 in the Supplementary Information).”

As an aside, in their ‘response to reviewers’, they state regarding their own manuscript (in contrast to prior work): ”In contrast, the main novel claim of the present paper is to show, in a generic way, that the negative damping (on average) effect of any pedestrian’s walking, at any stride frequency is essential for the pedestrian-induced instability. Moreover, when synchronisation occurs it is a consequence, rather than a cause, of the instability.”

This seems to be a statement (specifically the genericity claim) that may be hard to justify by the modeling or the mathematics. The authors’ critique of previous work (as a mathematical model not reality or having using a specific pedestrian model) in the response to reviewers is true of their manuscript as well – and indeed ALL modeling studies – that the results are contingent on the mathematical and structural assumptions embedded in the model. Even modeling the person’s effect as equivalent to the forces that they exert on the bridge based on a reduced state that consists of bridge and the person’s center of mass, say, is a remarkably simplifying assumption that has nothing generic about it. Generalizing foot force models do not take away from these assumptions. The genericity may be true for a particular mathematical model and particular types of model perturbations, but may not be true for all possible models for the phenomenon or model perturbations. To be clear, I don’t think there is anything wrong with such simplifying assumptions or lack of genericity, as long as grander claims are not made and the models are reasonable. And I *do* think the authors’ models are reasonable.

This is a good point, and we apologise for the loose language in our previous reply to the reviewer. Since this was only in the reply not the main text, we have not made a specific change (but see below for our response to the referee’s broader point about our bold claims in the manuscript). In our defense though (and indeed something we have made clear in the revised Introduction) the simulations of the three mathematical models provides only one part of the paper. In addition, we have produced a general asymptotic formula, a generic argument and an extensive review of the experimental literature. Please notice that we have included the following discussion of the limitations of our models in providing the full picture of possible confounding factors that can play a role in the initiation of bridge instability (red text starting from line 357). In particular, we noted that “Indeed, even when much is known about the physical properties of a bridge, knowledge of the crowd behaviour is necessarily subject to large uncertainties, both aleatoric and epistemic. For example, not only will there be a distribution of foot-placement control laws amongst the individuals in any crowd, but that distribution is not known” and ”Calibration of the models and inclusion of further features such as mode shapes and possible pedestrian-to-pedestrian interactions in dense crowds may lead to improved guidelines for bridge design.”

In any case, to repeat myself, I still feel it would be better if the authors were to tone down the claims to an appropriate level, more clearly and explicitly contrasting their work in the manuscript itself, and address the limitations of their model more clearly.

We have now done this by changing the title, the abstract, introduction and discussion (see red text). On reflection, we had previously got the tone wrong, and thank you for giving us one more opportunity to get it right.

The title ”A new paradigm for emergent instability: the real story of the London Millennium Bridge” still seems journalistic rather than scientifically descriptive. Further, while the authors have made some solid contributions, I do not know that their contributions warrant calling them ”the real story of the London Millennium Bridge” any more than the many other manuscripts they

have cited.

Thank you for the suggestion. Taking on board your broader point about toning down the rhetoric, the new title is "The London Millennium Bridge revisited: emergent instability without synchronisation". On the broader point of whether our paper has any more merit than other papers we have cited, we would nevertheless claim that our work is the most balanced and comprehensive to date (of course benefiting from hindsight) and makes the most compelling argument to date that the cause of the lateral pedestrian instability has consistently been footbridge excitation due to negative damping and not to crowd synchrony.

In my previous review, I had requested toning down such rhetoric but the authors argued that they still want to retain it. I would again suggest toning down such claims and make the title more descriptive, something about negative damping, phase-locking and synchrony, their main contributions as they themselves state. I do think these contributions are interesting.

We have toned down the rhetoric, as requested. On reflection, we got the balance wrong between making clear, strong statements and being dogmatic. We have made substantial changes to the abstract, introduction and discussion (see red text). Now, rather than bold claims about problems being solved and 'finally dispelling myths' we have changed the language to that of 'we argue strongly' and 'we provide overwhelming evidence that'.

Accessibility issues with the videos.

Four videos were attached in this submission. One of the videos was not play-able on regular quicktime player version 10.5 on my macbook computer: 2997411video5726949qw41wy.mp4 (but I was able to view it after converting it using ffmpeg, so my comments below are based on this conversion). Please check if your videos are open-able using standard software, or please suggest recommended software.

We have changed the format of this video to MKV, a universal format for TV shows and YouTube videos, so it should be playable on most devices and platforms.

Minor comments:

The 2997411video5726949qw41wy.mp4 shows the computer generated animation of one person walking. The authors have added a point-mass version of the animation, in response to my critique that the full-body model animation was misleading to the viewer. This is much better but the point-mass model animation has the potential issue that the legs do not connect at the body center of mass, but have a finite hip widths. Is this finite hip widths true of the mathematical model? Also, analogously, the complexity of the figures 1-2 are still misleading relative to the simplicity of the model. The authors could ideally explicitly note this in at least one of the figure captions (that the real model is much simpler) and/or have the point-mass model also depicted in the figures.

and pr Thank you for this suggestion. As requested, we have added the following clarification to the captions of Fig. 2: "The pedestrians are depicted as "crash test" dummies with flexible hips; however, the actual inverted pendulum model is simpler, with pendulum-like legs connecting to the CoM."

We have also added the following paragraph in red in the Methods (lines 519-523): "Note that the legs of the 3D humanoid avatar do not connect at the body centre of mass, but have a finite hip width. This hip width is not in the mathematical model. Only the CoM and CoP are modelled, with a rigid - though not necessarily direct straight - connection between them. The legs in the animation, though not drawn on a direct straight line between the CoM and CoP, connecting them rigidly, and the CoM and CoP lateral positions are exactly as found from the model."

Supplementary Videos: *In my previous review, I had expressed a number of concerns regarding the videos: specifically the un-controlled nature of the videos in Nepal and the fact that the human-induced aspect did not seem stark. Because of these concerns, I still believe that the videos do not add*

anything scientifically to the manuscript. Here are my concerns from the last version re-capped.

” - The authors present the video shot in Nepal as evidence for the lack of synchrony and in support of the model. However, there are multiple issues with using this video as evidence, even anecdotal evidence. First, one of the videos was shot by a person holding a camera while walking on the bridge, so its hard to disambiguate the cameraperson’s movement from the bridge movement; in the other two videos as well, there is considerable camera shaking, which makes it hard to observe bridge movement. To the extent that the bridge movement can be observed, a lot of it seems vertical in response to the pedestrians walking. There is a lot of wind (as can be seen from the fluttering flags), which makes it unclear if any of the small sideways bridge oscillations are due to wind or due to the pedestrians. ”

While the authors have added a good short paragraph in the Supplementary Information addressing the videos’s limitations, they should ideally state these reservations clearly in the main manuscript. Also, I’m not sure that statements like ”However, thousands of Everest-bound trekkers and climbers, 780 crossing the Hillary Bridge, named after Sir Edmund Hillary and featured in Fig. 7 b,c, would attest from their experience that ...” are admissible in a scientific article without either survey data or say a reference to a newspaper article that states something to the effect that is independent of the authors or these research). But reporting on personal experience sounds fine to me, as long as mentioned as an anecdote.

We have deleted the videos from the supplementary materials.

Reviewer 3:

The models are sound in themselves but they do not cover every possibility. They are fine for identifying the essentials of interaction between the pedestrian and the bridge. However, interactions between pedestrians is only through the behaviour of the bridge, and other possible interactions between the pedestrians, for example adaption necessary to accommodate each other’s movement or changes in visual field are excluded. As a result, this paper can only show that the proposed mechanism is the probable mechanism encapsulating the essence of the physics. By excluding other routes to coherence it cannot quite kill off the opposing view, although It does seem to kill off the Kuramoto route.

We have changed the wording to avoid being so dogmatic, to reflect the referee’s balanced assessment that this is a probable mechanism. In reality no theory can ever completely ”kill off the opposing view”. Please see the modifications to the abstract, introduction and discussion sections, shown in red. We have also added the following paragraph to further discuss the role of pedestrian-to-pedestrian interactions (lines 367-372): “Calibration of the models and inclusion of further features such as mode shapes and possible pedestrian-to-pedestrian interactions in dense crowds may lead to improved guidelines for bridge design. In particular, crowd congestion can cause footfall frequencies to enter into bands that are more likely to trigger instability [61], or human-to-human interactions may affect footstep timing. Our asymptotic formulae are well suited for addressing these research questions as the contribution of social force pedestrian dynamics [61] in promoting or damping instability can be explicitly evaluated via integral quantity σ_3 . These calculations are a subject of future work.”

Significance These results are likely to be significant in understanding the phenomenon in the field of pedestrian bridge design. However, the calculations of the scale and sign of the damping effect introduced by the pedestrians, and the consequent number of pedestrians to cause unstable oscillations, show a wide variation. The average values have a similar shape to the theoretical predictions, but the scale of the variation is significant. The authors are pessimistic as to how this might be used in design. The walking model is very basic and reality it may be significantly more complicated. The analysis also takes no account of the effects of the mode shape of the bridge’s vibration or variations in the individual mass, pace and distribution of walkers. These aspects, and

no doubt others, will affect the magnitude of the effect. However, the engineering profession is well adapted to using simple models. As long as the model captures the essence of the phenomenon it can be adapted and calibrated against measured data. I see no reason why this should not be possible here. In the end though, the only way out is to add structural damping, and all a more detailed analysis would give is a more precise figure about the level that is required.

The referee is right, there are many effects that would be required to inform a simple design criterion. At the request of another referee we have added an additional paragraph to the discussion (lines 351-365) that addresses precisely this point, and also the need for caution. In particular we make the important point that if the synchronisation hypothesis were used as a design criterion it could potentially be dangerous. Nevertheless, we hope that our work can inspire further empirical work by engineers to calibrate simple models with measured data, as the referee suggests.

Reviewers' Comments:

Reviewer #2:

Remarks to the Author:

1) I was unable to open the mkv videos on my computer with multiple video players without installing new software. I don't think this format is as universal as the authors claimed. As far as I can tell, neither windows media player (on windows) nor quicktime (on mac) opens the video. Looks like VLC is the only software that opens it across platforms. Was the editorial team at Nature Communications able to open it at their end?

2) I appreciate the authors revisiting the general tone of the manuscript and revising the title, abstract, and the main text of the manuscript. I very much think this measured tone reads much better and is appropriate for the results. I think the papers adds new results to this ongoing story of the London Millennium bridge. Thank you also for showing the actual changes in the revised manuscript/response, as it makes the reviewing easier.

3) The title of the manuscript now reads: "The London Millennium Bridge revisited: emergent instability without synchronisation". While this title is much better than the earlier title, the content of this title is quite similar to that of reference [17]: "Walking crowds on a shaky surface: stable walkers discover Millennium Bridge oscillations with and without pedestrian synchrony". Thus, as it stands, the new title seems (to me) to not be able to distinguish the main results of the two manuscripts. Perhaps some modification could be made to the title to make it a bit more specific. I do understand that the two manuscripts have different foci (see below).

4) The authors have indeed now described the main results of [17] in more detail. However, in their description, the authors state:
(Line 249) "However, once the stride frequencies were chosen to be equal, identical pedestrians triggered the bridge instability simultaneously with the onset of crowd synchrony".

But this description is not true of [17]. Indeed, if you look at Figures 2c and 2d of [17], we do see that there is persistent and growing oscillations even in the identical bipeds case, well before the bipeds synchronize. In Figure 2d, middle panel with 240 pedestrians, it takes about 100 time units to synchronize. But it is clear that there is persistent oscillations well below this moment of synchronization. Thus, it seems like the figures in [17] contains the result that synchrony is not necessary for substantial growth of the oscillations. Of course, as far as I can tell, the authors of [17] did not point this out, or focus on the relative timing of synchronization versus oscillation growth in identical bipeds. In any case, I encourage the authors to revise their description. And their most important contribution herein seems to be the perturbation analysis and derivation of new formulas that govern the bifurcation point.

5) From a mathematical perspective, the following sentence seems imprecise or at least informal or colloquial. Systems don't go stable or unstable. But particular equilibria/fixed points or other motions (invariant sets) go stable or unstable.
"At the macro-scale, the system goes unstable due to an analogue of fluid-structure interaction flutter, which in this case is actually a kind of non-smooth Hopf bifurcation."

6) The individual curves in the insets in Fig 3 and Fig 8 of SI of main manuscript are hard to discern. Also the curves in the inset seem to go beyond the extent of the inset, and thus seem truncated.

7) The authors are adding pedestrians one at a time at time intervals of $T_{\text{add}} = 20$ seconds. How was this parameter chosen? This seems a rather low frequency of person addition compared to real bridge walking. Does the instability persist when T_{add} is a) more realistic, and b) chosen arbitrarily? I am familiar with bifurcations that sometimes vanish during a rapid pass through the bifurcation point.

8) The authors show that there is a small region of no synchrony before there is synchrony. Is this entirely numerical result? Does the rate at which people are added affect the results?

Minor comment.

In the abstract, there is a remark that says "for any foot-force model". I'd suggest changing this to "for a class of foot-force models". The latter is true but the former is not true.

Reviewer #3:

Remarks to the Author:

Key Results

The authors present a convincing case for an alternative explanation to lateral vibrational instability, well-known from what happened to the Millennium Footbridge at the time of its opening. The classic explanation is that this behaviour is caused by pedestrians adjust their pacing to fall in step with each other at the natural frequency of the bridge. The authors present an impressive list of cases, garnered from around the world, where large induced lateral bridge motion has been observed and note that the evidence for synchrony in pacing is scant. More tellingly they describe cases where the synchrony argument cannot possibly be the cause of the motion because the bridge responds at frequencies well away from the pacing frequency. The authors' analysis shows that the prime effect of numbers of pedestrians walking over a bridge is to add, or in some cases remove, energy from the bridge. When enough pedestrians walk on the bridge the net effect is to overcome the bridge's own damping leading to the lateral vibrational instability.

Validity

The main findings are based on analyses of the interaction of increasing numbers of pedestrians placed on a simple one degree of freedom model of the bridge. Three pedestrian models are used. Two are based on the well accepted placement strategy due to Hof. The first assumes that the pedestrian walks at the same pace and the second is a new modified version that adjusts the step timing to take account of changes in gait due to bridge motion. The third is a model proposed by the authors and adopts a different strategy which has a greater dependence on the bridge motion and a greater tendency to synchronise. All these models are simple but show realistic behaviour. The simulations were primed with data to replicate the Millennium Footbridge around its opening and the results reproduce the observed behaviours of the bridge including the onset of unstable oscillations. It seems that the basic argument behind the paper is sound. Two methods of analysis have been used. The first is basically a Monte-Carlo simulation, which is a well-established technique, and the second is the application of a multiple scale asymptotic expansion, which produces similar results indicating support for the validity of the Monte-Carlo results.

Significance

These results are likely to be significant in understanding the phenomenon in the field of pedestrian bridge design. The authors do caution that the calculations of the scale and sign of the damping effect introduced by the pedestrians, and the consequent number of pedestrians to cause unstable oscillations, show a wide variation. They go on to point out the walking models are very basic and, in reality, they are likely to be significantly more complicated, adding that the analysis also takes no account of the effects of the mode shape of the bridge's vibration or variations in the individual mass, pace and distribution of walkers. All these aspects, and no doubt others, will affect the magnitude of the effect, so much so that, they say, the method is unlikely to be very helpful in detailed design of bridges. I think the authors are being a little too pessimistic here. Engineers are used to applying simple models to complicated cases and with more work the effect of complications the authors identify might be able to be assessed, possibly producing useful design guidance. However, the authors do point out that these results do sound a warning to designers now. It is shown that the reduction in damping does occur when the pedestrian pacing frequency and the natural frequency of the bridge are widely different, which not only supports the observations on the Clifton Bridge by Macdonald that other methods cannot explain, but also indicates that designers should not cut back on supplying additional damping when there is apparently little chance of synchrony.

Data and Methodology

The search for similar cases provides evidence that the problem is not confined to the Millennium Footbridge. The engineering industry is pretty good at reporting odd cases, particularly where

there is public concern in the behaviour of a structure, and especially since the case of the Millennium Footbridge highlighted the phenomenon, so it is likely cases of large bridge oscillations get reported. It is

The appropriate literature is well reviewed and discussed. The choice of models is reasonable presenting a simple and a more complicated 'standard' Hof model, and a third with a significantly different control algorithm. The analysis is essentially a Monte Carlo approach, which is a well-established method. The multiple scale asymptotic method provides a significant support

Analytical approach

The simulation models are reasonable, and the algorithms used to solve the equations are bread and butter stuff so can be considered to produce very reliable results. The Monte Carlo method is a well-worn approach and will produce good estimates of the mean behaviour. I am less able to judge the validity of the multiple scale analysis. I can follow the maths, which seems logical and reasonable, but it is a bit out of my field so I cannot really provide any valuable comment here.

Clarity and context

The paper is well written in lucid English. The argument is clearly stated and follows a good logical line. It is now over 20 years since the completion of the Millennium footbridge and the events surrounding its opening. Over the intervening period there has been much discussion and, more importantly, research into the cause. This paper does outline the beginnings of a method that might assess pedestrian induced lateral motion correctly. Given that it has taken so long to get to what looks like the true explanation indicates that some of the criticism heaped on the 'naive' designers is not justified. This paper does set that straight, but more importantly in engineering terms is that the original engineers did fix the problem pretty quickly.

References

The paper does reference previous literature appropriately.

Expertise

I have a background in structural dynamics and testing, and experience of human/structure interaction.

Reply to Reviewer 2

1) I was unable to open the mkv videos on my computer with multiple video players without installing new software. I don't think this format is as universal as the authors claimed. As far as I can tell, neither windows media player (on windows) nor quicktime (on mac) opens the video. Looks like VLC is the only software that opens it across platforms. Was the editorial team at Nature Communications able to open it at their end?

Reply: We were able to open this video on various PC and MAC computers. To respond to the reviewer's concern, we are also submitting this video in an alternative format.

2) I appreciate the authors revisiting the general tone of the manuscript and revising the title, abstract, and the main text of the manuscript. I very much think this measured tone reads much better and is appropriate for the results. I think the papers adds new results to this ongoing story of the London Millennium bridge. Thank you also for showing the actual changes in the revised manuscript/response, as it makes the reviewing easier.

Reply: We are thankful to the reviewer for these remarks and positive assessment of the changes made.

3) The title of the manuscript now reads: "The London Millennium Bridge revisited: emergent instability without synchronisation". While this title is much better than the earlier title, the content of this title is quite similar to that of reference [17]: "Walking crowds on a shaky surface: stable walkers discover Millennium Bridge oscillations with and without pedestrian synchrony". Thus, as it stands, the new title seems (to me) to not be able to distinguish the main results of the two manuscripts. Perhaps some modification could be made to the title to make it a bit more specific. I do understand that the two manuscripts have different foci (see below).

Reply: We agree that there is an overlap with the titles; however, our work focuses on synchronization being a consequence, not a cause, of instability. We were unable to come up with a better title, so we let the reader appreciate the differences between the foci of the two papers.

4) The authors have indeed now described the main results of [17] in more detail. However, in their description, the authors state: (Line 249) "However, once the stride frequencies were chosen to be equal, identical pedestrians triggered the bridge instability simultaneously with the onset of crowd synchrony".

But this description is not true of [17]. Indeed, if you look at Figures 2c and 2d of [17], we do see that there is persistent and growing oscillations even in the identical bipeds case, well before the bipeds synchronize. In Figure 2d, middle panel with 240 pedestrians, it takes about 100 time units to synchronize. But it is clear that there is persistent oscillations well below this moment of synchronization. Thus, it seems like the figures in [17] contains the result that synchrony is not necessary for substantial growth of the oscillations. Of course, as far as I can tell, the authors of [17] did not point this out, or focus on the relative timing of synchronization versus oscillation growth in identical bipeds. In any case, I encourage the authors to revise their description. And their most important contribution herein seems to be the perturbation analysis and derivation of new formulas that govern the bifurcation point.

Reply: We apologise for the confusion. As in our current manuscript, we compared the onset of instability and of synchronisation in terms of the related (different) crowd sizes. Fig. 2d in [17] compares the transients for a fixed crowd size and, therefore, points out to a different phenomenon. However, as the reviewer rightfully mentions, [17] did not explicitly focus on the relative timing of synchronization versus oscillation growth in identical bipeds. To avoid confusions, we have deleted this passage from the paper. The updated description is as follows:

"In particular, this work ([17]) used an energy-optimised pedestrian model with a linear feedback controller to demonstrate that heterogeneous pedestrians incapable of synchronising even at large

crowd sizes can shake the bridge without synchronisation [17].”

To be fair to the authors of Barker [28] and Macdonald [18] who previously studied this effect using different foot force models, we have also added the following sentence: “This effect was also reported in an earlier paper by Baker [28] and described for Model 1 in Macdonald [18].”

5) From a mathematical perspective, the following sentence seems imprecise or at least informal or colloquial. Systems don’t go stable or unstable. But particular equilibria/fixed points or other motions (invariant sets) go stable or unstable. ”At the macro-scale, the system goes unstable due to an analogue of fluid-structure interaction flutter, which in this case is actually a kind of non-smooth Hopf bifurcation.”

Reply: This kind of construction is of common usage in nonlinear science.

6) The individual curves in the insets in Fig 3 and Fig 8 of SI of main manuscript are hard to discern. Also the curves in the inset seem to go beyond the extent of the inset, and thus seem truncated.

Reply: We agree that these individual curves are hard to distinguish; however, the main point of demonstrating them is to visualise the transition from uncorrelated to correlated pedestrian motions. These transitions are clearly seen in the figures.

7) The authors are adding pedestrians one at a time at time intervals of $T_{add} = 20$ seconds. How was this parameter chosen? This seems a rather low frequency of person addition compared to real bridge walking. Does the instability persist when T_{add} is a) more realistic, and b) chosen arbitrarily? I am familiar with bifurcations that sometimes vanish during a rapid pass through the bifurcation point.

Reply: This is a good point. To clarify the choice of T_{add} , we have added the following description:

“We performed our simulations for two different choices of pedestrian addition times $T_{add} = 20$ s and $T_{add} = 10$ s. These choices are consistent with the incremental pedestrian loading tests on the London Millennium Bridge [21] and simulations conducted by Ingolfsson et al. [64] in which pedestrians were added at average intervals of 7 s and 12 s, respectively.”

Please notice that since the original submission, there has been separate subsection “Faster addition of pedestrians to the bridge” in the Supplementary Information. This section contains the results for $T_{add} = 10$ and addresses the reviewer’s point as follows: “In this case, the pedestrian-bridge system has a narrower time window for transient effects before the addition of the next pedestrian. As a result, one can expect that the crowd will have grown larger by the time the vibrations have increased in amplitude significantly. The simulations displayed in Supplementary Fig. 1 confirm this intuition and indicate the widening of the instability region (pink) preceding the onset of weak (Model 2) and strong synchronisation (Model 3).”

To better point out the reader to this section, we have added the following paragraph: “Representative results for $T_{add} = 20$ s are depicted in Fig. 3, with further results in the Supplementary Information (see Supplementary Fig. 1 for faster pedestrian addition time $T_{add} = 10$ s and Supplementary Fig. 2 for the worst-case scenario of complete resonance).”

8) The authors show that there is a small region of no synchrony before there is synchrony. Is this entirely numerical result? Does the rate at which people are added affect the results?

Reply: This point was addressed in the previous revision. As mentioned in section “Simulation results,” the calculations of negative damping are based on the asymptotic formula, whereas the calculations of the order parameter are entirely numerical. As mentioned above, the role of the rate at which pedestrians are added is discussed in the Supplementary Information.

Minor comment. In the abstract, there is a remark that says “for any foot-force model”. I’d suggest changing this to ”for a class of foot-force models”. The latter is true but the former is not true.

Reply: We have change “any foot-force model” to “a wide class of foot-force models.”

Reply to Reviewer 3

Reviewer 3 (Remarks to the Author):

Key Results The authors present a convincing case for an alternative explanation to lateral vibrational instability, well-known from what happened to the Millennium Footbridge at the time of its opening. The classic explanation is that this behaviour is caused by pedestrians adjust their pacing to fall in step with each other at the natural frequency of the bridge. The authors present an impressive list of cases, garnered from around the world, where large induced lateral bridge motion has been observed and note that the evidence for synchrony in pacing is scant. More tellingly they describe cases where the synchrony argument cannot possibly be the cause of the motion because the bridge responds at frequencies well away from the pacing frequency. The authors’ analysis shows that the prime effect of numbers of pedestrians walking over a bridge is to add, or in some cases remove, energy from the bridge. When enough pedestrians walk on the bridge the nett effect is to overcome the bridge’s own damping leading to the lateral vibrational instability. **Validity** The main findings are based on analyses of the interaction of increasing numbers of pedestrians placed on a simple one degree of freedom model of the bridge. Three pedestrian models are used. Two are based on the well accepted placement strategy due to Hof. The first assumes that the pedestrian walks at the same pace and the second is a new modified version that adjusts the step timing to take account of changes in gait due to bridge motion. The third is a model proposed by the authors and adopts a different strategy which has a greater dependence on the bridge motion and a greater tendency to synchronise. All these models are simple but show realistic behaviour. The simulations were primed with data to replicate the Millennium Footbridge around its opening and the results reproduce the observed behaviours of the bridge including the onset of unstable oscillations. It seems that the basic argument behind the paper is sound. Two methods of analysis have been used. The first is basically a Monte-Carlo simulation, which is a well-established technique, and the second is the application of a multiple scale asymptotic expansion, which produces similar results indicating support for the validity of the Monte-Carlo results. **Significance** These results are likely to be significant in understanding the phenomenon in the field of pedestrian bridge design. The authors do caution that the calculations of the scale and sign of the damping effect introduced by the pedestrians, and the consequent number of pedestrians to cause unstable oscillations, show a wide variation. They go on to point out he walking models are very basic and, in reality, they are likely to be significantly more complicated, adding that the analysis also takes no account of the effects of the mode shape of the bridge’s vibration or variations in the individual mass, pace and distribution of walkers. All these aspects, and no doubt others, will affect the magnitude of the effect, so much so that, they say, the method is unlikely to be very helpful in detailed design of bridges. I think the authors are being a little too pessimistic here. Engineers are used to applying simple models to complicated cases and with more work the effect of complications the authors identify might be able to be assessed, possibly producing useful design guidance. However, the authors do point out that these results do sound a warning to designers now. It is shown that the reduction in damping does can occur when the pedestrian pacing frequency and the natural frequency of the bridge are widely different, which not only supports the observations on the Clifton Bridge by Macdonald that other methods cannot explain, but also indicates that designers should not cut back on supplying additional damping when there is apparently little chance of synchrony. **Data and Methodology** The search for similar cases provides evidence that the problem is not

confined to the Millennium Footbridge. The engineering industry is pretty good at reporting odd cases, particularly where there is public concern in the behaviour of a structure, and especially since the case of the Millennium Footbridge highlighted the phenomenon, so it is likely cases of large bridge oscillations get reported. The appropriate literature is well reviewed and discussed. The choice of models is reasonable presenting a simple and a more complicated 'standard' Hof model, and a third with a significantly different control algorithm. The analysis is essentially a Monte Carlo approach, which is a well-established method. The multiple scale asymptotic method provides a significant support Analytical approach The simulation models are reasonable, and the algorithms use to solve the equations are bread and butter stuff so can be considered to produce very reliable results. The Monto Carlo method is a well-worn approach and will produce good estimates of the mean behaviour. I am less able to judge the validity of the multiple scale analysis. I can follow the maths, which seems logical and reasonable, but it is a bit out of my field so I cannot really provide any valuable comment here. Clarity and context The paper is well written in lucid English. The argument is clearly stated and follows a good logical line. It is now over 20 years since the completion of the Millennium footbridge and the events surrounding its opening. Over the intervening period there has been much discussion and, more importantly, research into the cause. This paper does outline the beginnings of a method that might assess pedestrian induced lateral motion correctly. Given that it has taken so long to get to what looks like the true explanation indicates that some of the criticism heaped on the 'naive' designers is not justified. This paper does set that straight, but more importantly in engineering terms is that the original engineers did fix the problem pretty quickly. References The paper does reference previous literature appropriately. Expertise I have a background in structural dynamics and testing, and experience of human/structure interaction.

Reply: We are grateful to the reviewer for the positive assessment of our work.